# Balancing Gradient Frequencies Facilitates Inductive Inference in Algorithmic Reasoning

## Abstract

Inductive inference, or extrapolation of general rules from finite instances, is understood to be the foundation of human intelligence. Unfortunately, Deep Neural Networks (DNNs) struggle with inductive inference and thus fail to learn even the simplest algorithms in Algorithmic Reasoning (AR). Existing research efforts on AR with DNNs are limited to those on the architectural design for DNNs. In this study, we investigate the influence of optimization techniques on AR performance. Through toy experiments designed to understand an optimizer's susceptibility to shortcuts in AR, we reveal that Adam, the naïve choice of optimization, is easily fooled by spurious correlations. To overcome this shortcoming of Adam, we propose a novel optimizer that avoids spurious correlations by balancing gradients of low- and high-frequencies (BGF). We present extensive experiments and analyses to demonstrate the broad and multifaceted advantages of BGF across various architectures and AR tasks. In particular, BGF expands the AR capability of all explored DNN models and even shows the potential to enable learning of tasks that they previously failed at. The observed success of BGF in climbing the Chomsky hierarchy underscores the importance of optimization for developing advanced artificial intelligence with DNNs.

## 1 Introduction

The goal of Algorithmic Reasoning (AR) (Veličković & Blundell, 2021), expressed through sequence prediction tasks, is to learn the general output-generating logic from the finite data with a maximum sentence length of $l \leq N$, which can be generalized to arbitrary length $l \in [1, \infty)$. Therefore, the ability to perform inductive inference, which involves deriving a general rule from finite instances, is the key to the successful execution of AR (Delétang et al., 2022). While inductive inference is understood to be the foundation of human-level intelligence (Pazzani & Kibler, 1992; Klauer & Phye, 2008), Deep Neural Networks (DNNs), the most powerful of current-day machine learning (ML) models, are incapable of conducting inductive inference. For instance, Large Language Models (LLMs) with remarkable human-like emergent properties (Brown et al., 2020; Chen et al., 2021; Chowdhery et al., 2023; Touvron et al., 2023) are known to struggle with learning simple algorithms like addition or parity through inductive inference (Nogueira et al., 2021; Ontanon et al., 2021; Wu et al., 2023; Saparov et al., 2024; Anil et al., 2022).

DNNs' struggle with inductive inference is attributed to the lack of statistical guarantee for their out-of-distribution (OOD) generalization property. The current generalization property of ML models, including DNNs, is grounded on the statistical learning theory (Vapnik et al., 1998), which assumes that training and test data are independent and identically distributed and does not consider OOD scenarios where test data do not belong in the training distribution (Solomonoff, 1964; Fisher, 1935). Searching for the desired OOD generalization solution among many hypotheses fitting the training data requires meta-information outside given training data (Mitchell, 1980). In the DNN training process, this meta-information is translated into inductive bias, governed by the DNN architecture, optimization methods, and initial parameters (Goyal & Bengio, 2022). Previously, extensive research has been conducted on how various DNN architectures can perform different types of logical reasoning tasks (Skachkova et al., 2018; Chomsky, 1956; Merrill, 2019). Research on Transformers reported that while they can learn some Dyck languages (Bhattamishra et al., 2020), in most AR tasks, they tend to learn statistically brittle shortcuts and fail to learn the underlying algorithms (Liu et al., 2022; Lee et al., 2023; Abbe et al., 2023; Nogueira et al., 2021). In Deletang *et al.* (Delétang et al., 2022)

and Merrill *et al.* (Merrill et al., 2022), comprehensive experiments are conducted to investigate the ability of recurrent models to learn AR tasks of varying difficulties (Chomsky, 1956).

Contrary to the exhaustive studies on architectures, the influence of optimization methods on OOD generalization in the context of AR has been largely overlooked. Through carefully designed toy experiments, we analyze whether Adam (Kingma & Ba, 2014), the de facto optimizer in AR research, models the general output-generating logic. In our toy experiments, we create synthetic spurious correlations between train inputs and ground truth outputs. These spurious correlations function as easily tractable proxies to conceptually abstract shortcuts and enable explicit analysis of AR tasks from an optimization standpoint. Our exploratory study reveals that Adam exhibits inherent vulnerability to shortcuts and necessitates advancements on the optimization front to achieve true OOD generalization.

In this regard, we propose a novel optimizer, named BGF (Balancing of Gradient Frequencies), which circumvents shortcuts and enhances OOD generalization by balancing gradients of high- and low-frequencies. We first observe the occurrence of "train-test splitting," where OOD generalization on test data is achieved far after the model fits training data, when training DNNs on AR tasks with Adam. Train-test splitting can be interpreted as a mild form of grokking (Power et al., 2022). A frequency-domain analysis of gradients obtained over the course of AR training indicates that high-frequency components, which are dominant at early training steps where the training accuracy increases, are visibly suppressed once the training accuracy saturates. Following the suppression of high-frequency components, the test accuracy starts to improve, implying that low-frequency components provide a necessary diversion from shortcuts to guide the training process to learn the general underlying logic. Thus, we posit that balancing high- and low-frequency gradients can improve OOD generalization. Our initial results of replacing Adam with BGF in our toy experiments consolidate that BGF is indeed more capable of evading shortcuts, consolidating its capability to encourage learning of general rules.

Extensive experimental results evidence that BGF improves AR performance and accelerates AR learning on eight DNN models employed by Deletang *et al.* (Delétang et al., 2022). 1) BGF improves the best test accuracy of evaluated models on all 15 AR tasks and shows potential to enable learning of high-complexity tasks that they previously failed at (*i.e.*, Recurrent Neural Network (RNN) on the Missing Duplicate task). Length generalization in AR was perceived to be bounded by the inductive bias of a DNN architecture. BGF's model-agnostic evasion of shortcuts relieves this strong contingency of AR performance on the choice of a DNN architecture. 2) BGF alleviates train-test splitting and reduces the number of training steps required to obtain over $90\%$ accuracy on test data.

In addition, we present the results of diverse analyses that uncover the advantages of BGF. The comparison of BGF with optimizers proposed to improve generalization in conventional ML illuminates that BGF is a superior method of optimization for AR. The verification of BGF's generalization capability on longer sequences shows that BGF has facilitated learning of the underlying logic. Analyzing the variance of the training speeds computed over different initializations corroborates BGF's capability to stabilize the AR training process. The sharpness analysis of the loss landscapes obtained with BGF shows that it smoothens the loss surface of trained DNNs, offering insights into how BGF induces DNNs with improved generalization capacity on AR. Our contributions are summarized as follows:

- We propose BGF, a novel optimizer that is capable of improving length generalization in AR through the mitigation of shortcut learning. Notably, BGF shows an instance of climbing the Chomsky hierarchy by enabling the complete execution of a previously impossible task.

- To accentuate the necessity of an optimization technique tailored toward AR, we disclose the vulnerability of Adam to shortcuts in AR tasks through toy experiments. In contrast, the verification of BGF on these toy experiments shows its strong promise at battling shortcuts.

- The existence of the train-test splitting phenomenon in AR is reported for the first time. Analyzing the change in gradient patterns over AR training steps reveals that when the test accuracy starts to improve, the high-frequency components in gradients are visibly suppressed; this analysis inspires the design of BGF that balances low- and high-frequencies.

- Our extensive experimental results demonstrate that BGF brings upon over-arching improvements in performance and training speed. We provide insightful analyses into what makes BGF advantageous for length generalization in AR.

## 2 PRELIMINARY

### 2.1 ALGORITHMIC REASONING AND CHOMSKY HIERARCHY

While DNNs excel at function approximation and feature extraction (Mao & Jain, 1995), they exhibit unpredictable behavior beyond the confines of the training data (Vapnik, 1999; Liang et al., 2018). Conversely, classical algorithms, grounded in predetermined rules, offer scalability to diverse datasets, performance guarantees, and interpretability. The complementary nature of these methodologies inspired numerous studies on equipping DNNs with the characteristics of classical algorithms through algorithmic modeling (Trask et al., 2018; Yan et al., 2020). However, these endeavors are limited by their reliance on defining task-specific parameterized functions under human supervision (Gaunt et al., 2017; Valkov et al., 2018). The emerging "learning from data" approach, which involves acquiring proficiency in arbitrary logical tasks without task supervision, holds promise for addressing this limitation (Delétang et al., 2022; Liu et al., 2022; Veličković et al., 2022; Minder et al., 2023).

AR refers to the process of learning underlying algorithms through finite pairs of input and output sequence data (train length $l < N$). The inference is then conducted on unobserved inputs to determine whether the model has learned an underlying rule capable of generalization (inference length $l >> N$). There exist two main approaches to representing AR tasks, sequence prediction (Delétang et al., 2022) and graph learning (Veličković et al., 2022). In this paper, we employ the sequence prediction approach because it theoretically subsumes all computational problems and can be represented with formal languages (Sipser, 1997; Rich, 2007).

The Chomsky hierarchy organizes formal languages into four levels according to their complexity. Regular languages (R) and corresponding finite automata require the simplest memory complexity. Context-free (CF) languages and push-down automata necessitate stack-type memory. Context-sensitive (CS) languages and linear bounded automata require linear tape memory. Lastly, recursively enumerable (RE) languages and Turing machine demand infinite tape memory. The levels of formal language (in which AR tasks are expressed) that various DNN models can learn revealed the Chomsky hierarchy boundary for each architecture, as summarized in Table A2 (Delétang et al., 2022).

### 2.2 SPURIOUS CORRELATION AND OUT-OF-DISTIBUTION GENERALIZATION

In AR, DNNs models, despite achieving 100% accuracy on training data, fail to generalize to test cases. This lack of generalization capability, particularly in Transformer models, is commonly associated with their tendency to learn easier shortcuts that apply only to the training data rather than general rules. While preventing models from learning shortcuts or spurious correlations and enhancing OOD generalization performance are central goals in machine learning, limited research has been conducted for AR tasks from these perspectives.

Methods to mitigate spurious correlations can be broadly categorized into data manipulation, representation learning, and learning strategy (Liu et al., 2021b; Ye et al., 2024). Approaches in data manipulation include corrupting semantic information in the data (Puli et al., 2022), increasing diversity through mix-up (Yao et al., 2022), using counterfactual generators (Zeng et al., 2020), and pseudo-label generation (Nam et al., 2022). Even though data manipulation is demonstrated to be effective in vision and natural language tasks, its application to AR is implausible since the distribution of training inputs in AR tasks is numerically defined and spans all possible input spaces.

Representation learning, whose aim is to learn domain-agnostic robust representations, has been widely adopted to enable OOD generalization (Zhang et al., 2022; Harary et al., 2022). Unsupervised representation learning approach is unsuitable for AR tasks as it only relies on input patterns. For instance, parity check and binary addition share identical input domains, requiring output information for learning. For supervised representation learning, methods with environment labels (Pfister et al., 2019; Albuquerque et al., 2019; Zhu et al., 2023) also do not align with the philosophy of AR. Methods that do not rely on environment labels commonly decorrelate input into label-relevant and irrelevant information (Zhang et al., 2021; Lachapelle et al., 2023; Creager et al., 2021; Liu et al., 2021a). These approaches are also impractical for AR, as every pixel of the input data is associated with labels, and a single pixel change can alter the class.

The limitations of data manipulation and representation learning leave us with optimization-based mitigation of spurious correlations. Fortunately, loss landscape sharpness-based optimization strate-

gies have shown promise in simple logical tasks (Klindt, 2022). The correlation between the loss landscape sharpness and generalization has been studied theoretically (Hochreiter & Schmidhuber, 1994) and practically (Dinh et al., 2017). Optimization methods enforcing loss landscape flatness include LPF-SGD (Bisla et al., 2022) and SAM (Foret et al., 2021). An alternative approach emulates this effect through weight ensembling (Izmailov et al., 2018; Cha et al., 2021). SWAD (Cha et al., 2021) employs a dense, overfit-aware weight sampling strategy for ensemble selection.

## 3 METHODOLOGY

Existing research efforts in AR are mostly directed toward studying the influence of different DNN architectures and their inductive bias on OOD generalization performance. However, the design of the optimizer, another important dimension that constitutes inductive bias, remains underexplored in AR. In this study, we aim to investigate and improve OOD generalization in AR tasks from the perspective of optimization. Through the two toy experiments in Section 3.1, we disclose the susceptibility of Adam, the default optimizer used in AR, to shortcuts, embodied in the form of spurious correlations. In Section 3.2, we observe the occurrence of train-test splitting in AR and reveal that low-frequency components in gradients contribute to the increase in test, instead of training, accuracy. Inspired by this analysis, we propose BGF, a novel optimizer that circumvents shortcuts and encourages the learning of generic rules in train data, by balancing of high- and low-frequency gradients.

### 3.1 MOTIVATION

**Definition of Shortcuts:** In AR, the Adam optimizer typically achieves near-perfect training accuracy, exhausting available information from the data, but yields noticeably lower test accuracy. Thus, AR distinguishes itself from typical ML problems where in distribution (ID) accuracy often predicts or correlates with OOD generalization (Miller et al., 2021; Hendrycks et al., 2021) and requires performing inductive inference. The main challenge in inductive inference (including AR) is that data alone cannot determine a single hypothesis among those that fit the data. For example, predicting the next number in the sequence 1, 3, 5, 7 has infinite possible continuations. In this work, among hypotheses fitting the training data, those that enable OOD generalization (e.g., 9 for the continuation) are defined as "general logic", whereas non-generalizing hypotheses are termed "shortcuts". Shortcuts can be interpreted as learning non-generalizing correlations between inputs and outputs that are valid in the training distribution but do not hold in the target distribution; such misleading correlations are regarded to be spurious correlations.

In this section, we expose that the vulnerability of Adam to the shortcuts in AR tasks creates a major roadblock for DNNs trying to learn the underlying data-generating logic of AR. Unfortunately, it is infeasible for us humans to pinpoint and explicate which shortcuts are implicitly learned by models. Therefore, we design toy experiments as a test bed to study the susceptibility of current optimization approaches to shortcuts. In these toy experiments, we utilize synthetic spurious correlations between train inputs and outputs as representations of conceptually abstract shortcuts. Comparing the robustness of different optimizers to spurious correlations allows close monitoring of whether an optimizer can avoid shortcuts embodied in the form of tangible confounding factors.

We study two different types of spurious correlations, whose examples for the "Modular Arithmetic" task are depicted in Figure 1. (The **Modular Arithmetic (MA)** task is defined to be: given a sequence of inputs and operations sampled from $\{0, 1, 2, 3, 4\}$ and $\{+, -, \times\}$, compute modulo 5 of the result of applying the operation on the sequence.) The first spurious correlation is induced by concatenating the ground truth output to the original input sequence as seen in Figure 1(b). At test time, the ground truth output is replaced with a randomized value to observe whether the optimizer learned the logic embedded in the original input without memorizing the concatenated ground truth. The second type of synthetic confounding factor we utilize is the input length. The example in Figure 1(c) depicts how the "Modular Arithmetic" task is modified to implement input length confounding. In this task, the potential input length ranges from 1 to 40, and the number of ground truth outputs is 5. Consequently, our confounding rule partitions data into five groups according to the input length: the inputs of length $1 \sim 8$ are labeled as 0, those of length $9 \sim 16$ as 1, and so forth. To force spurious correlation based on the input length, we only select instances that satisfy the above rule and discard the rest. The model trained on manipulated train data is tested on normal test data sans spurious correlation. In both toy experiments, the degree of spurious correlation is controlled with probability $P$. For instance,

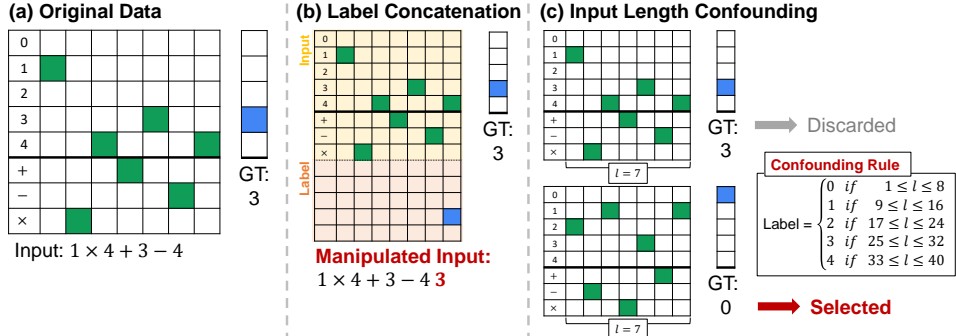

Figure 1: (a) depicts an example of the original "Modular Arithmetic" task. In (b), the ground truth output is attached to the input sequence, creating the opportunity for the DNN to memorize this ground truth instead of learning the arithmetic rule. In (c), we present a confounding logic to create a spurious correlation between the input length and the ground truth output. In the given example, when the length of the example input is 7, we only select examples with the ground truth output of 0.

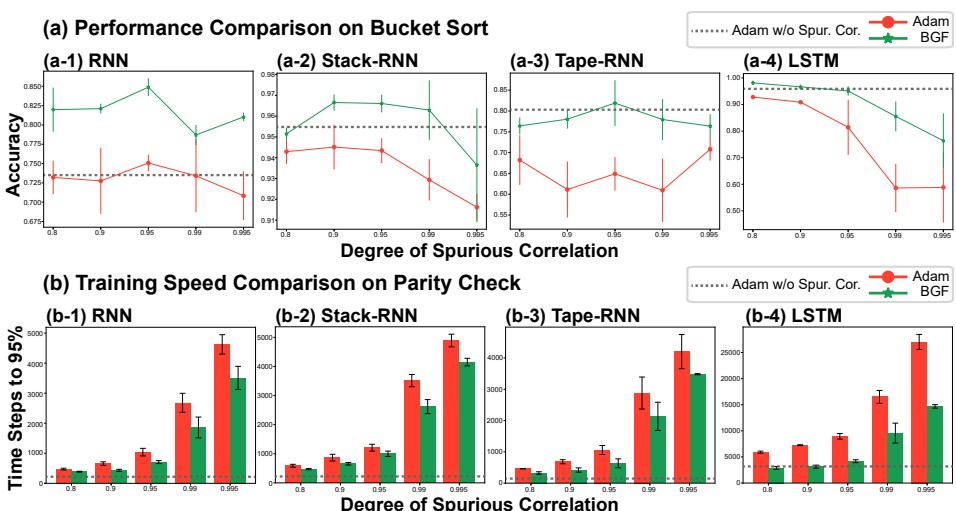

Figure 2: Qualitative visualization of how spurious correlation with label concatenation affects the performance and the training speed of Adam (Red) and BGF (Green). Adam shows a decrease in performance and an increase in training speed as stronger degrees of spurious correlations are induced. BGF exhibits visible improvements over Adam on both fronts.

when $P = 0.8$, $80\%$ of train data are manipulated to exhibit spurious correlation. The length of the train and test input sequences are set to be $1 \sim 40$ and $100$, respectively. All the results of our toy experiments are averages of results obtained over three different seeds. The first toy experiment is conducted on eleven tasks, in which the input length is greater than or equal to the output length. This condition is necessary to concatenate the ground truth without changing the length of the original input sequence. In Figure 2, we visualize the effect of this synthetic spurious correlation on the performance and training speed of Adam, quantified through its best test accuracy and the number of time steps it took to reach $95\%$ accuracy, respectively. In both figures, the dotted gray line denotes the original performance and training speed of Adam obtained from clean training data free of synthetic spurious correlations ($P = 0$). Adam experiences a consistent drop in performance or substantial increase in training steps with the introduction of spurious correlations of various degrees.

The second experiment is conducted on seven tasks with an output length of one because it is ambiguous to create length-based spurious correlations on tasks with an output length greater than one. Due to the page limit, the extended results of both toy experiments are included in Appendix A3. In congruence with the qualitative results from the first toy experiment, Adam shows a consistent decline in performance when spurious correlations are introduced. These results collectively demonstrate that Adam is susceptible to learning spurious correlations in AR tasks, highlighting the need for a new optimizer better suited for training DNN models on such tasks.

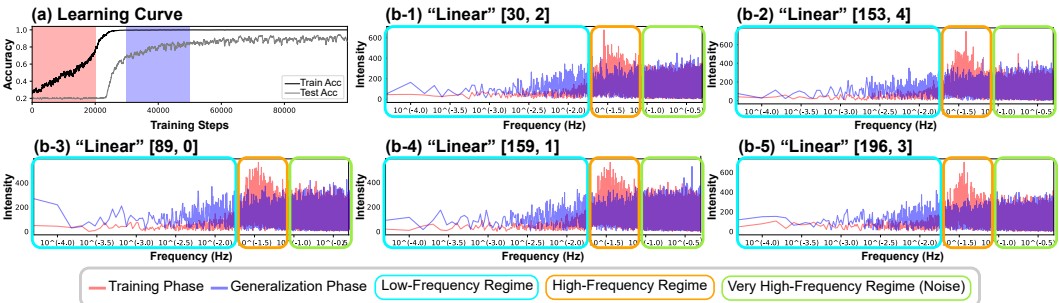

Figure 3: (a) shows how training and test accuracies evolve during training. (b-1 ∼ 5) visualize FFT results of normalized gradient signals from five randomly sampled layers. The title for each plot notes the layer from which the gradient was obtained. Following (a), red and blue signals correspond to FFT results of training and generalization phase gradients, respectively. Cyan, orange, and green boxes mark low-, high-, and very-high frequency regions. The intensity of the generalization phase gradient is visibly dominant in the low-frequency regime.

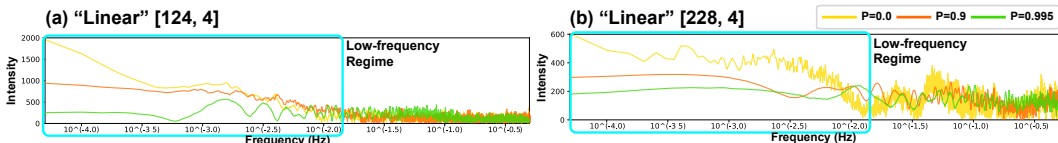

Figure 4: (a) and (b) visualize FFT results of normalized gradients from the label concatenation toy experiment on Bucket Sort. The title for each plot notes the layer from which the gradient was obtained. Gradients from the first 20,000 training steps are collected to analyze the impact of spurious correlations. Training gradients from P=0.0 (yellow), 0.99 (orange), and 0.995 (green) are compared.

## 3.2   BGF: MITIGATING SHORTCUT LEARNING VIA BALANCING OF GRADIENT FREQUENCIES

We take a closer look at the AR training process of Adam by plotting the training and test accuracy curves of the LSTM model on the "Solve Equation" task in Figure 3 (a). Training and test accuracies improve at two disparate points in the training curve. This phenomenon, which we dub "train-test splitting," is a moderate version of grokking. From here on, we denote the point in the learning curve in which training accuracy increases but test accuracy does not as the training phase (red), and the point in which only test accuracy increases as the generalization phase (blue).

To understand what learning signals contribute to improving the test accuracy and enabling OOD generalization, we analyze the evolution of gradients over the course of training. Figure 3 (b 1 ∼ 5) are the Fast Fourier Transform (FFT) results of gradient signals obtained from training and generalization phases. In early steps, when the training accuracy continuously rises while the test accuracy is stagnant, the gradient patterns are dominated by high-frequency components (red signal). Interestingly, the intensity of high-frequency gradients substantially attenuates in later gradient signals that yield OOD generalization capability (blue signal), further validating that relatively lower frequency gradients are preferable for learning of the general logic. Such a result indicates that the low-frequency components play a crucial role in avoiding shortcuts and promoting the learning of general logic.

We additionally analyze the normalized gradient signals from the "Bucket Sort" task with different degrees of spurious correlation induced by label concatenation. The FFT results for gradient signals obtained at $P \in [0.0, 0.99, 0.995]$ are visualized in Figure 4. As the degree of spurious correlation between input data and labels gets stronger, the low-frequency components of gradient signals visibly decrease. This result further evidences that in AR, spurious correlations or shortcuts significantly weaken low-frequency gradients, and that amplifying low-frequency gradients may hold the key to facilitating inductive inference.

These analyses lead us to the design of BGF, a novel optimizer that guides the training process toward general logic and away from shortcuts by balancing the gradients in low- and high-frequency regimes. The most intuitive approach to obtaining low-frequency gradients is through the adoption of a low-pass filter, which removes high-frequency components with stochastic and fine-grained signals

in gradients of a single training step. Instead of converting gradients to the frequency domain, BGF mimics the behavior of a low-pass filter by applying the moving average filter with the window size of $\lambda$. $\lambda$ is used to adjust the filtering range in a low-pass filter and is treated as a hyperparameter that controls the cut-off frequency. Let $g_t$ denote the gradient of some loss function $\mathcal{L}$ computed with respect to the trainable parameters $\theta_t$ at $t$-th training step: $g_t = \nabla_{\theta_t} \mathcal{L}(\theta_t)$. We store gradients from $t - 1 - \lambda$ to $t - 1$ training steps in a gradient queue $\mathcal{Q}_{\text{grad}}$, which has a fixed size of $\lambda$ by definition: $\mathcal{Q}_{\text{grad}} = \{g_i\}_{i=t-1-\lambda}^{i=t-1}$. The past gradients are averaged over the time dimension to compute the low-pass filtered gradients: $g_{\text{low}_t} = \frac{1}{\lambda} \sum_{i=t-1-\lambda}^{i=t-1} g_i$.

Once the low-pass filtered gradients have been obtained, they are balanced with the high-frequency gradients computed from the current batch as follows:

$$g_{\text{BGF}_t} = \alpha * g_t + \frac{\beta * ||g_t||_F}{||g_{\text{low}_t}||_F} * g_{\text{low}_t}, \quad \text{where } \alpha + \beta = 1.0. \tag{1}$$

$|| \cdot ||_F$ denotes the Frobenius norm computed on each layer of a DNN model. $\alpha$ and $\beta$ are coefficients used to balance two gradient signals. We normalize the low-pass filtered gradients to stabilize the optimization process of BGF by aligning the scale of two disparate gradients. We report the results of removing the balancing and normalization steps of BGF in Appendix A4; they show that this normalization step is crucial for stabilizing BGF.

The initial results of replacing Adam with BGF in the toy experiments from Section 3.1 demonstrate that BGF is more competent than Adam at avoiding spurious correlations. On the first toy experiment, Figure 2 delineates that BGF surpasses the performance of Adam and accelerates the training speed across all levels of spurious correlation. The extended quantitative comparison of the performance of Adam and BGF on both toy experiments, presented in Appendix A3, further consolidates the effectiveness of BGF.

## 4 RESULTS

### 4.1 EXPERIMENTAL SET-UP

**Tasks and Data Generation:** We utilize the 15 sequence prediction tasks defined in Deletang *et al.* for empirical verification of our method. The list of tasks, associated abbreviations, and detailed formalization can be found in Appendix A1. The synthetic data for each task are generated following the procedure in Deletang *et al.*. The ranges of training and test sequence lengths are $1 \sim 40$ and $100$, respectively. All training processes are executed for $100,000$ steps.

**Architectures:** To demonstrate BGF can be utilized in an architecture-agnostic manner, we consider eight different DNN architectures: 4 variants of a recurrent model (RNN, Stack-RNN (Joulin & Mikolov, 2015), Tape-RNN (Suzgun et al., 2019; Delétang et al., 2022), LSTM (Hochreiter & Schmidhuber, 1997b)) and 4 variants of the Transformer encoder, each one with different positional encoding: none (None), classical sin/cos (SinCos) (Vaswani et al., 2017), ALiBi (ALIBI) (Press et al., 2021), and relative positional encodings of Transformer-XL (Rot) (Dai et al., 2019).

**Implementation and Training Details:** All our experiments are implemented using JAX (Bradbury et al., 2018) and PyTorch (Paszke et al., 2019) and run on NVIDIA A40 and L40 GPUs. More details on hyperparameters used for compared methods are included in Appendix A2. Unless stated otherwise, all of the reported results are averages of three different random seeds.

### 4.2 PERFORMANCE COMPARISON

In Table 1, we compare the best test accuracy obtained by Adam and BGF on 120 task - architecture combinations (15 tasks $\times$ 8 architectures). BGF's superiority to Adam is clearly demonstrated through performance improvements in 93.3% of combinations, with an average increase in accuracy of 3.3%p (p-value $< 0.001$). Notably, BGF exhibits staggering improvement of 46.1%p on the "Missing Duplicate" task (MD) with Tape-RNN, and there are 11 scenarios where BGF's improvement exceeds 10%p. We further emphasize that on the "MD" task, training RNN with BGF achieves 100% test accuracy. This impressive result indicates that BGF enables learning of the MD task with RNN, which represents an instance of climbing the Chomsky hierarchy. In Appendix A5, we compare the performance of AdamW (Loshchilov, 2017), a more advanced variant of Adam, and BGF on

Table 1: Best accuracy comparison against Adam, a naïve approach to optimization in AR. Rows corresponding to the performance of BGF are highlighted in blue for improved visibility. Accuracies over 90%, considered to be indications of achieving OOD generalization, are marked in bold.

| Level | Task | Optim. | RNN | Stack-RNN | Tape-RNN | LSTM | TF (None) | TF (Sin-Cos) | TF (ALIBI) | TF (Rot) |
|---|---|---|---|---|---|---|---|---|---|---|
| R | EP | Adam | **1.000** | **1.000** | **1.000** | **1.000** | 0.531 | 0.629 | **0.960** | **1.000** |
| | | BGF | **1.000** | **1.000** | **1.000** | **1.000** | 0.533 | 0.683 | **0.967** | **1.000** |
| | MA | Adam | **1.000** | **1.000** | **1.000** | **1.000** | 0.223 | 0.222 | 0.227 | 0.227 |
| | | BGF | **1.000** | **1.000** | **1.000** | **1.000** | 0.227 | 0.223 | 0.252 | 0.248 |
| | PC | Adam | **1.000** | **1.000** | **1.000** | **1.000** | 0.529 | 0.529 | 0.529 | 0.529 |
| | | BGF | **1.000** | **1.000** | **1.000** | **1.000** | 0.556 | 0.533 | 0.532 | 0.541 |
| | CN | Adam | **1.000** | **1.000** | **1.000** | **0.978** | 0.861 | 0.271 | 0.590 | 0.494 |
| | | BGF | **1.000** | **1.000** | **1.000** | **1.000** | **0.979** | 0.349 | 0.871 | 0.566 |
| CF | SM | Adam | 0.572 | 0.763 | 0.793 | 0.712 | 0.507 | 0.518 | 0.613 | 0.597 |
| | | BGF | 0.577 | 0.763 | 0.826 | 0.718 | 0.510 | 0.530 | 0.624 | 0.605 |
| | RS | Adam | 0.743 | 0.713 | 0.722 | 0.716 | 0.542 | 0.539 | 0.699 | 0.719 |
| | | BGF | 0.761 | 0.715 | 0.731 | 0.707 | 0.542 | 0.542 | 0.728 | 0.708 |
| | MAB | Adam | 0.458 | 0.796 | 0.508 | 0.820 | 0.326 | 0.327 | 0.326 | 0.321 |
| | | BGF | 0.512 | **0.987** | **0.947** | **0.912** | 0.327 | 0.327 | 0.328 | 0.326 |
| | SE | Adam | 0.844 | **0.908** | 0.859 | **0.962** | 0.222 | 0.225 | 0.226 | 0.228 |
| | | BGF | 0.843 | **0.906** | 0.896 | **0.953** | 0.224 | 0.226 | 0.225 | 0.223 |
| CS | DS | Adam | 0.541 | 0.558 | 0.541 | 0.650 | 0.541 | 0.536 | 0.534 | 0.535 |
| | | BGF | 0.546 | 0.584 | 0.548 | 0.696 | 0.542 | 0.541 | 0.539 | 0.578 |
| | MD | Adam | 0.544 | 0.635 | 0.539 | 0.560 | 0.549 | 0.545 | 0.603 | 0.560 |
| | | BGF | **1.000** | 0.725 | **1.000** | 0.706 | 0.589 | 0.550 | 0.656 | 0.592 |
| | OF | Adam | 0.544 | 0.575 | 0.541 | 0.560 | 0.542 | 0.539 | 0.536 | 0.564 |
| | | BGF | 0.548 | 0.593 | 0.558 | 0.684 | 0.542 | 0.541 | 0.539 | 0.563 |
| | BA | Adam | 0.496 | 0.534 | 0.507 | 0.596 | 0.510 | 0.506 | 0.502 | 0.536 |
| | | BGF | 0.504 | 0.545 | 0.515 | 0.634 | 0.513 | 0.513 | 0.506 | 0.558 |
| | BM | Adam | 0.503 | 0.530 | 0.502 | 0.551 | 0.500 | 0.504 | 0.503 | 0.554 |
| | | BGF | 0.506 | 0.530 | 0.508 | 0.578 | 0.501 | 0.506 | 0.506 | 0.554 |
| | CS | Adam | 0.569 | 0.609 | 0.569 | 0.618 | 0.513 | 0.513 | 0.511 | 0.577 |
| | | BGF | 0.573 | 0.612 | 0.601 | 0.628 | 0.514 | 0.514 | 0.516 | 0.581 |
| | BS | Adam | 0.738 | **0.954** | 0.805 | **0.957** | 0.255 | 0.391 | 0.700 | **0.948** |
| | | BGF | 0.867 | **0.980** | 0.844 | **0.989** | 0.255 | 0.501 | **0.924** | **0.959** |

recurrent models. These results further demonstrate that BGF clearly outperforms AdamW (with an average increase of 3.3%p and a maximum increase of 46.8%p) and successfully learns tasks that AdamW failed to learn completely. To demonstrate that the performance discrepancy between Adam and BGF is not a byproduct of the difference in their ability to fit the training distribution, in Appendix A6, we present the average "training distribution accuracy" associated with Table 1. Both Adam and BGF achieve near-perfect accuracy on the training distribution (the average accuracy of Adam is 0.979, and that of BGF is 0.983), underscoring that the AR tasks require inductive inference. Notably, for the "MD" task, the training distribution accuracy exceeds 0.999 in all cases.

### 4.3 Training Speed Comparison

In this section, we compare the training speed of Adam and BGF in terms of the number of time steps each optimizer takes to reach (a) 90% and (b) 95% accuracy. For each AR task, we average the training speed over all architectures with the exception of architectures on which neither one of the optimizers achieves 90% or 95% accuracy. For visualization purposes, we plot $0.5 - T_{\mathrm{BGF}}/(T_{\mathrm{Adam}} + T_{\mathrm{BGF}})$, where

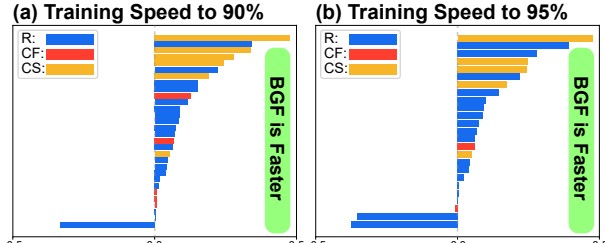

Figure 5: Visualization of training step comparison.

$T_{\mathrm{Adam}}$ and $T_{\mathrm{BGF}}$ denote the training speed of Adam and BGF measured in time steps, respectively. When this value is greater than 0, it means that BGF achieves the target accuracy with fewer steps than Adam. The results in Figure 5 demonstrate that BGF reduces the number of training steps to reach 90% or 95% accuracy in the majority of AR tasks. BGF's convergence speed is particularly superior to Adam "context-free" and "context-sensitive" tasks, which are more challenging than "Regular" tasks.

Table 2: Best accuracy comparison against existing optimizers borrowed from conventional ML.

| Architecture | Optimizer | Train | | | | | | Test | | | | | |
|---|---|---|---|---|---|---|---|---|---|---|---|---|---|
| | | SM | RS | SE | MD | BA | BS | SM | RS | SE | MD | BA | BS |
| **RNN** | Adam | 0.988 | 1.000 | 0.999 | 1.000 | 0.893 | 0.995 | 0.572 | 0.743 | 0.844 | 0.544 | 0.496 | 0.738 |
| | SAM | 0.941 | 0.998 | 0.905 | 0.997 | 0.799 | 0.965 | 0.568 | 0.707 | 0.465 | 1.000 | 0.490 | 0.430 |
| | SWAD | 0.975 | 1.000 | 0.908 | 1.000 | 0.855 | 0.994 | 0.561 | 0.716 | 0.222 | 0.540 | 0.492 | 0.751 |
| | LPF-SGD | 0.972 | 1.000 | 0.934 | 1.000 | 0.854 | 0.994 | 0.557 | 0.715 | 0.225 | 0.543 | 0.493 | 0.722 |
| | BGF | 0.996 | 1.000 | 0.999 | 1.000 | 0.911 | 0.999 | 0.577 | 0.761 | 0.843 | 1.000 | 0.504 | 0.867 |
| **Stack-RNN** | Adam | 1.000 | 1.000 | 1.000 | 1.000 | 0.943 | 1.000 | 0.763 | 0.713 | 0.908 | 0.635 | 0.534 | 0.954 |
| | SAM | 1.000 | 1.000 | 0.998 | 0.999 | 0.874 | 0.997 | 0.766 | 0.709 | 0.746 | 0.724 | 0.507 | 0.495 |
| | SWAD | 1.000 | 1.000 | 0.985 | 1.000 | 0.886 | 1.000 | 0.762 | 0.714 | 0.444 | 0.567 | 0.509 | 0.914 |
| | LPF-SGD | 1.000 | 1.000 | 0.989 | 1.000 | 0.880 | 1.000 | 0.763 | 0.713 | 0.760 | 0.625 | 0.512 | 0.915 |
| | BGF | 1.000 | 1.000 | 1.000 | 1.000 | 0.953 | 1.000 | 0.763 | 0.715 | 0.906 | 0.725 | 0.545 | 0.980 |
| **Tape-RNN** | Adam | 1.000 | 1.000 | 1.000 | 1.000 | 0.959 | 1.000 | 0.793 | 0.722 | 0.859 | 0.539 | 0.507 | 0.805 |
| | SAM | 1.000 | 1.000 | 0.993 | 0.996 | 0.865 | 0.998 | 0.713 | 0.657 | 0.645 | 0.547 | 0.492 | 0.507 |
| | SWAD | 1.000 | 1.000 | 1.000 | 1.000 | 0.892 | 0.994 | 0.806 | 0.693 | 0.760 | 0.537 | 0.501 | 0.784 |
| | LPF-SGD | 1.000 | 1.000 | 0.992 | 1.000 | 0.889 | 0.995 | 0.754 | 0.675 | 0.757 | 0.535 | 0.509 | 0.762 |
| | BGF | 1.000 | 1.000 | 1.000 | 1.000 | 0.962 | 1.000 | 0.826 | 0.731 | 0.896 | 1.000 | 0.515 | 0.844 |
| **LSTM** | Adam | 1.000 | 1.000 | 1.000 | 1.000 | 1.000 | 1.000 | 0.712 | 0.716 | 0.962 | 0.560 | 0.596 | 0.957 |
| | SAM | 0.995 | 0.999 | 0.995 | 1.000 | 0.930 | 1.000 | 0.617 | 0.684 | 0.738 | 0.565 | 0.512 | 0.781 |
| | SWAD | 0.996 | 1.000 | 0.985 | 1.000 | 0.912 | 1.000 | 0.640 | 0.674 | 0.224 | 0.552 | 0.503 | 0.892 |
| | LPF-SGD | 0.997 | 1.000 | 0.988 | 1.000 | 0.911 | 1.000 | 0.638 | 0.662 | 0.221 | 0.552 | 0.499 | 0.888 |
| | BGF | 1.000 | 1.000 | 1.000 | 1.000 | 0.999 | 1.000 | 0.718 | 0.707 | 0.953 | 0.706 | 0.634 | 0.989 |

# 5 ANALYSES

## 5.1 COMPARISON WITH OTHER OPTIMIZERS

We now compare our approach against three additional optimizers with a shared goal of evading spurious correlations: SAM (Foret et al., 2021), SWAD (Cha et al., 2021), and LPF-SGD (Bisla et al., 2022). The description and hyperparameter settings of each optimizer are in Appendix A2. In Table 2, we compare the train and test accuracies of five optimizers. The extended results are in Appendix A7. Surprisingly, the previously proposed optimizers are completely inept at improving the OOD generalization performance on test sequences despite attaining $\sim 100\%$ on training distribution. The failure of optimizers adopted from conventional ML indicates that seeking generalization alone does not automatically lead to improved logic-based length generalization in AR.

## 5.2 EXCLUSIVENESS AND FLEXIBILITY OF BGF

Table 3: Experiments are conducted on the "MD" task. [Left] shows Adam and BGF with various momentum parameters trained for 1 million steps. [Right] shows results for Adam, BGF, and BGF implemented with an exponential moving average (BGF$_{\text{ema}}$).

| Arch | 1M steps | | | | | | Arch | 100k steps | | |
|---|---|---|---|---|---|---|---|---|---|---|
| | Adam$_{0.8}$ | BGF$_{0.8}$ | Adam$_{0.9}$ | BGF$_{0.9}$ | Adam$_{0.95}$ | BGF$_{0.95}$ | | Adam | BGF | BGF$_{\text{ema}}$ |
| RNN | 0.528 | **1.000** | 0.528 | **1.000** | 0.530 | **1.000** | RNN | 0.544 | **1.000** | **1.000** |
| Stack-RNN | 0.675 | 0.628 | 0.684 | 0.651 | 0.758 | 0.667 | Stack-RNN | 0.635 | 0.725 | **1.000** |
| Tape-RNN | 0.529 | **1.000** | 0.529 | **1.000** | 0.530 | **0.993** | Tape-RNN | 0.539 | **1.000** | **1.000** |
| LSTM | 0.625 | 0.701 | 0.549 | 0.743 | 0.656 | 0.650 | LSTM | 0.560 | 0.706 | 0.582 |

**Comparison to Single Momentum:** Here, we observe how changing the momentum parameter in Adam affects the performance of Adam and BGF. In Table 3 [Left], we compare the performance of BGF against that of Adam with three different momentum parameters (0.8, 0.9, 0.95). These results are obtained after 1 million training steps to show that increasing the number of training steps does not bridge the performance gap between the two optimizers. Across all momentum strengths, BGF outperforms Adam, and Adam consistently fails to achieve OOD generalization. This experiment brings to light that because $g_{\text{BGF}}$ balances low- and high-frequency gradients, it is infeasible to substitute $g_{\text{BGF}}$ and mimic its effects with single momentum.

**BGF Implementation with Exponential Moving Average (EMA):** We also study the effect of implementing the gradient filtering in BGF with an EMA instead of a queue. The results of BGF with EMA in Table 3 [Right] show that implementing BGF with EMA instead of a queue achieves OOD generalization on the "MD" task with reduced memory cost. In particular, we note that BGF with EMA reaches 100% accuracy additionally on Stack-RNN.

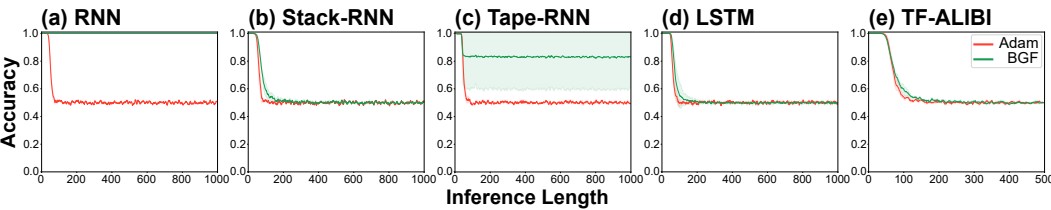

Figure 6: Performance comparison of Adam *vs.* BGF on test sequences of longer lengths.

Table 5: Loss landscape analysis of the DNN models trained with Adam and BGF.

| Measure | Regular | | | Context-Free | | | Context-Sensitive | | |
|---|---|---|---|---|---|---|---|---|---|
| | LPF ($\downarrow$) | Fim ($\downarrow$) | Shannon ($\downarrow$) | LPF ($\downarrow$) | Fim ($\downarrow$) | Shannon ($\downarrow$) | LPF ($\downarrow$) | Fim ($\downarrow$) | Shannon ($\downarrow$) |
| Adam | 0.3712 | 0.0206 | 1.08e-04 | 0.0581 | 363.34 | 0.0107 | 0.5305 | 3810.23 | 0.1047 |
| BGF | 0.1027 | 0.0053 | 5.61e-05 | 0.0405 | 315.23 | 0.0122 | 0.3175 | 3353.29 | 0.0937 |
| p-value | 0.129 | 0.220 | 0.237 | 0.205 | 0.372 | 0.394 | 0.182 | 0.345 | 0.015 |

## 5.3 ADVANTAGEOUS PROPERTIES OF BGF

**Extreme Length Generalization:** To test whether BGF truly learns the general logic of the MD task, traditionally considered to be unlearnable by RNNs, we verify its length generalization capability on test sequences of lengths up to 1000. Figure 6 demonstrates that the RNN trained with BGF maintains its performance even under this extreme OOD scenario. Results on additional tasks are provided in Appendix A8. This result supports that BGF induces learning of the underlying algorithm.

**Variance Reduction:** We analyze the variance of training speed to demonstrate that BGF is not only faster but also more stable at training DNNs on AR tasks. Like in Section 4.3, cases where neither Adam nor BGF achieves 90% accuracy are omitted from comparison. In cases where only a subset of the three seeds reach 90% accuracy, we calculate the variance assuming a Poisson distribution. Table 4 shows that across all DNN models, BGF exhibits lower variance in training speed, which can be equated to improved training stability.

Table 4: Standard deviation of the training speed of Adam and BGF and associated p-value of the paired t-test.

| | RNN ($\downarrow$) | SR ($\downarrow$) | TR ($\downarrow$) | LSTM ($\downarrow$) | TF ($\downarrow$) |
|---|---|---|---|---|---|
| Adam | 27.4k | 28.4k | 28.0k | 30.2k | 39.9k |
| BGF | 6.0k | 16.6k | 20.5k | 22.4k | 33.8k |
| p-value | 0.106 | 0.130 | 0.256 | 0.015 | 0.733 |

**Sharpness Analysis**: It is widely accepted that the generalization performance of a model is negatively correlated with the sharpness of its loss landscape (Hochreiter & Schmidhuber, 1997a). We compare Adam and BGF from the optimization landscape perspectives using three representative measures of sharpness: low-pass filter-based method (LPF) (Bisla et al., 2022), the Fisher Rao Norm-based method (Fim) (Liang et al., 2019), and the Shannon Entropy-based method (Shannon) (Pereyra et al., 2016). The results in Table 5 provide a meaningful insight that the superior performance of BGF is a product of its smoother loss landscape.

## 6 CONCLUSION

To the best of our knowledge, this is the first work to study the effect of optimization on generalization in AR. Our investigative results exposed the susceptibility of Adam to spurious correlations in AR tasks, necessitating the development of a new optimizer. We then analyzed the evolution of gradients throughout the AR training process, unveiling the occurrence of the train-test splitting phenomenon and the contribution of low-frequency components to enabling OOD generalization. Based on these findings, we proposed BGF, a novel optimizer to encourage learning of general logic in AR, and demonstrated its effectivenss through extensive experiments and analyses. **Limitations:** Although BGF considerably improves the AR performance compared to Adam, it still fails to promote learning of many AR tasks. Therefore, a considerable amount of research opportunities remain on both the architecture design and optimization approach fronts to enable DNNs to perform inductive inference.

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

# A APPENDIX

## A1 15 SEQUENCE PREDICTION TASKS FOR ALGORITHMIC REASONING

Table A1: Summary of AR tasks each DNN model is traditionally known to be able to perform. † and ★ denote permutation-invariant and counting tasks; ○ refers to tasks that demand a nondeterministic controller; and × indicates the requirement of super linear running time in terms of the input length. The checkmark denotes that the DNN model can perform the task in the corresponding row.

| Level | Task | RNN | Stack-RNN | Tape-RNN | Transformer | LSTM |
|-------|------|-----|-----------|----------|-------------|------|
| R | Even Pairs (EP) | ✓ | ✓ | ✓ | ✓ | ✓ |
| | Modular Arithmetic (MA) | ✓ | ✓ | ✓ | | ✓ |
| | Parity Check† (PC) | ✓ | ✓ | ✓ | | ✓ |
| | Cycle Navigation† (CN) | ✓ | ✓ | ✓ | | ✓ |
| CF | Stack Manipulation (SM) | | ✓ | ✓ | | |
| | Reverse String (RS) | | ✓ | ✓ | | |
| | Modular Arithmetic Brackets (MAB) | | ✓ | ✓ | | |
| | Solve Equation° (SE) | | | | | |
| CS | Duplicate String (DS) | | | ✓ | | |
| | Missing Duplicate (MD) | | | ✓ | | |
| | Odds First (OF) | | | ✓ | | |
| | Binary Addition (BA) | | | ✓ | | |
| | Binary Multiplication× (BM) | | | | | |
| | Compute Sqrt (CS) | | | | | |
| | Bucket Sort†★ (BS) | | | | ✓ | ✓ |

Table A2: AR tasks categorized according to the Chomsky hierarchy and associated example input/output pairs. This table was largely borrowed from (Delétang et al., 2022).

| Level | Task | Example Input | Example Output |
|-------|------|---------------|----------------|
| R | Even Pairs (EP) | *aaabbaabba* | True |
| | Modular Arithmetic (MA) | $1 \times 4 + 3 - 4$ | 3 |
| | Parity Check† (PC) | *abbbbaa* | True |
| | Cycle Navigation† (CN) | *010211* | 2 |
| CF | Stack Manipulation (SM) | *abaab* POP PUSH *a* POP | *abaa* |
| | Reverse String (RS) | *abaab* | *baaba* |
| | Modular Arithmetic Brackets (MAB) | $1 \times (4 + 3) - 4$ | 3 |
| | Solve Equation° (SE) | $1 \times (4 + x) - 4$ | 0 |
| CS | Duplicate String (DS) | *abaab* | *abaababaab* |
| | Missing Duplicate (MD) | 10011021 | 0 |
| | Odds First (OF) | *abaab* | *aabba* |
| | Binary Addition (BA) | $1010 + 11$ | 1101 |
| | Binary Multiplication× (BM) | $1010 \times 11$ | 11110 |
| | Compute Sqrt (CS) | 10000 | 100 |
| | Bucket Sort†★ (BS) | 431401 | 011344 |

Our empirical verification is conducted on 15 sequence prediction tasks proposed by Delétang et al. (2022). For the completeness of our work, we include a summary of each task below. We clearly acknowledge that this summarized description of task formulation is largely borrowed from the original paper (Delétang et al., 2022).

**Regular tasks**
- **Even Pairs (EP)**: Computer the number of a's and b's in a binary sequence, *e.g.*, babaa.
- **Modular Arithmetic (MA)**: Given a sequence of inputs and operations sampled from $\{0, 1, 2, 3, 4\}$ and $\{+, -, \times\}$, compute modulo 5 of the result of applying the operation on the sequence.
- **Parity Check (PC)**: Given a binarized input sequence, *e.g.,* aaabba, compute if the number of b's is even.
- **Cycle Navigation (CN)**: Based on a sequence of movements with the cycle length 5, compute

the final position. The potential movements are {STAY, INCREASE, DECREASE}, which are represented as: {0, 1, 2}.

**Context Free tasks**
- **Modular Arithmetic Brackets (MAB)**: This task is the same as the Modular Arithmetic Task with the exception of brackets.
- **Reverse String (RS)**: Determine the reverse of a given binary string.
- **Solve Equation (SE)**: Given equation that is constructed by numbers from {0, 1, 2, 3, 4}, brackets, and operations from {+, −, ×}, and an unknown variable, solve the equation to compute the value of z such that the equation holds after modulo 5.
- **Stack Manipulation (SM)**: Based on a binary string representation of a stack's contents (ordered by: bottom-to-top) and an action sequence constructed by a combination of potential actions {PUSH a, PUSH b, POP}, determine the final stack content after applying the action sequence on the stack.

**Context Sensitive tasks**
- **Binary Addition (BA)**: Compute the sum in base 2 of two binary numbers.
- **Binary Multiplication (BM)**: Compute the product in base 2 of two binary numbers.
- **Compute Sqrt (CS)**: Compute the floor of the square root of a binary number.
- **Duplicate String (DS)**: Output the original binary string and its duplicate.
- **Missing Duplicate (MD)**: The input consists of a binary string and its duplicate. Find one token that has been masked from the original binary string.
- **Odds First (OF)**: Output the values at the odd positions of the binary string first, followed by those at the even positions.
- **Bucket Sort (BS)**: Given an input sequence over an alphabet of fixed size, sort this string in an ascending order.

## A2 TRAINING DETAILS FOR BGF AND OTHER BASELINES

In this section, we provide descriptions and hyperparameter settings of compared optimizers in our main experiments. As noted in the main paper, all training processes are run for $100,000$ steps.
- **Adam**: is the default optimizer used in (Delétang et al., 2022)
- **SAM** (Foret et al., 2021): searches for neighboring regions with uniformly low losses. To do so, it applies perturbations to gradients during training, so that they optimize losses in the neighboring regions as well. The degree of perturbation in SAM is set to $0.5$. The perturbed gradients are multiplied by this value perturbation to control their effect.
- **LPF-SGD** (Bisla et al., 2022): reduces the influence of the data and label noise by applying the low pass filter kernel to the loss. The integral in the convolution operation is approximated with the MC method.
- **SWAD** (Cha et al., 2021): determines the training interval in which overfitting has occurred by tracking the validation loss. It then averages all parameters in this training interval to obtain the final model. The interval for parameter saving in SWAD is determined by three values: an optimum patient parameter ($N_p$), an overfitting patient parameter ($N_o$), and a tolerance rate $r_{tol}$. The start saving step happens when the test loss no longer decreases for $N_p$ number of steps. The end step happens when the test loss value exceeds $r_{tol}$ for $N_o$ number of steps. All parameters within this interval are saved. $N_p$, $N_o$, and $r_{tol}$ are set to 10, and 10, and 0.005, such that a reasonable interval is selected.
- **BGF (Ours)**: $\alpha$ and $\beta$ are two hyperparameters involved in BGF. Their sum is always controlled to be one, so that BGF balances their influences. $\alpha$ and $\beta$ are searched by conducting a hyperparameter search on the following sets of $\{\alpha, \beta\}$: $\{0.95, 0.05\}$, $\{0.9, 0.1\}$,$\{0.8, 0.2\}$,$\{0.7, 0.3\}$.

## A3 COMPREHENSIVE RESULTS ON TOY EXPERIMENTS

Table A3 and Table A4 present the full results of the Adam and BGF performance in the Label Concatenation experiment (Toy 1), respectively. Table A5 and Table A6 showcase the full results of the Adam and BGF performance in the Input Length Confounding experiment (Toy 2), respectively. In Table A3 and Table A5, we report the best test accuracy achieved during 100,000 steps of training.

For Tables Table A4 and Table A6, we report the training step at which the test accuracy first reached 90%. All values are averaged across three random seeds. For seeds that did not achieve 90% accuracy throughout the entire training, we report the performance at 100,000 steps and calculate the average. In the Label Concatenation experiment, we did not conduct experiments when the best accuracy of both Adam and BGF with the synthetic spurious correlation probability $P$ 0 is close to random ($<$ 0.3 for MA, CN, MAB, SE, BS, and $<$ 0.6 for the rest). These cells are left blank.

Table A3: Full results of the best accuracy in the Label Concatenation toy experiment. We excluded tasks from the experiment where both Adam and BGF resulted in outcomes close to random chance and left these cells blank.

| Level | Task | Method | RNN | | | | | Stack-RNN | | | | | Tape-RNN | | | | | LSTM | | | | |
|---|---|---|---|---|---|---|---|---|---|---|---|---|---|---|---|---|---|---|---|---|---|---|
| | | | 0.8 | 0.9 | 0.95 | 0.99 | 0.995 | 0.8 | 0.9 | 0.95 | 0.99 | 0.995 | 0.8 | 0.9 | 0.95 | 0.99 | 0.995 | 0.8 | 0.9 | 0.95 | 0.99 | 0.995 |
| R | EP | NNCH | 1.000 | 1.000 | 1.000 | 1.000 | 1.000 | 1.000 | 1.000 | 1.000 | 1.000 | 1.000 | 1.000 | 1.000 | 1.000 | 1.000 | 1.000 | 1.000 | 1.000 | 1.000 | 1.000 | 1.000 |
| | | Ours | 1.000 | 1.000 | 1.000 | 1.000 | 1.000 | 1.000 | 1.000 | 1.000 | 1.000 | 1.000 | 1.000 | 1.000 | 1.000 | 1.000 | 1.000 | 1.000 | 1.000 | 1.000 | 1.000 | 1.000 |
| | MA | NNCH | 1.000 | 1.000 | 1.000 | 1.000 | 0.996 | 1.000 | 1.000 | 1.000 | 1.000 | 0.959 | 1.000 | 1.000 | 1.000 | 1.000 | 0.807 | 1.000 | 1.000 | 1.000 | 0.997 | 0.221 |
| | | Ours | 1.000 | 1.000 | 1.000 | 1.000 | 0.857 | 1.000 | 1.000 | 1.000 | 0.996 | 0.222 | 1.000 | 1.000 | 1.000 | 0.999 | 0.651 | 1.000 | 1.000 | 1.000 | 1.000 | 0.505 |
| | PC | NNCH | 1.000 | 1.000 | 1.000 | 1.000 | 1.000 | 1.000 | 1.000 | 1.000 | 1.000 | 1.000 | 1.000 | 1.000 | 1.000 | 1.000 | 1.000 | 1.000 | 1.000 | 1.000 | 1.000 | 1.000 |
| | | Ours | 1.000 | 1.000 | 1.000 | 1.000 | 1.000 | 1.000 | 1.000 | 1.000 | 1.000 | 1.000 | 1.000 | 1.000 | 1.000 | 1.000 | 1.000 | 1.000 | 1.000 | 1.000 | 1.000 | 1.000 |
| | CN | NNCH | 1.000 | 1.000 | 1.000 | 1.000 | 1.000 | 1.000 | 1.000 | 1.000 | 1.000 | 1.000 | 1.000 | 1.000 | 1.000 | 1.000 | 1.000 | 1.000 | 1.000 | 1.000 | 1.000 | 0.962 |
| | | Ours | 1.000 | 1.000 | 1.000 | 1.000 | 1.000 | 1.000 | 1.000 | 1.000 | 1.000 | 1.000 | 1.000 | 1.000 | 1.000 | 1.000 | 1.000 | 1.000 | 1.000 | 1.000 | 1.000 | 1.000 |
| CF | RS | NNCH | 0.711 | 0.717 | 0.717 | 0.713 | 0.714 | 0.710 | 0.710 | 0.709 | 0.712 | 0.713 | 0.610 | 0.593 | 0.662 | 0.592 | 0.615 | 0.703 | 0.699 | 0.636 | 0.572 | 0.548 |
| | | Ours | 0.723 | 0.725 | 0.714 | 0.718 | 0.719 | 0.712 | 0.708 | 0.708 | 0.708 | 0.710 | 0.662 | 0.659 | 0.709 | 0.635 | 0.658 | 0.686 | 0.660 | 0.630 | 0.589 | 0.565 |
| | MAB | NNCH | 0.332 | 0.307 | 0.283 | 0.256 | 0.239 | 0.514 | 0.394 | 0.284 | 0.255 | 0.236 | 0.329 | 0.304 | 0.266 | 0.260 | 0.249 | 0.711 | 0.611 | 0.422 | 0.291 | 0.291 |
| | | Ours | 0.338 | 0.310 | 0.287 | 0.250 | 0.231 | 0.397 | 0.286 | 0.268 | 0.255 | 0.238 | 0.427 | 0.268 | 0.267 | 0.246 | 0.231 | 0.826 | 0.679 | 0.600 | 0.306 | 0.291 |
| | SE | NNCH | 0.852 | 0.769 | 0.497 | 0.219 | 0.219 | 0.819 | 0.781 | 0.657 | 0.219 | 0.219 | 0.841 | 0.782 | 0.607 | 0.219 | 0.219 | 0.891 | 0.856 | 0.726 | 0.219 | 0.219 |
| | | Ours | 0.811 | 0.695 | 0.430 | 0.219 | 0.219 | 0.849 | 0.798 | 0.712 | 0.219 | 0.219 | 0.827 | 0.735 | 0.458 | 0.219 | 0.219 | 0.931 | 0.880 | 0.791 | 0.219 | 0.219 |
| | MD | NNCH | 0.686 | 0.841 | 0.857 | 1.000 | 1.000 | 0.590 | 0.632 | 0.614 | 1.000 | 1.000 | 0.525 | 0.526 | 0.841 | 1.000 | 1.000 | 0.526 | 0.528 | 0.529 | 0.528 | 0.527 |
| | | Ours | 1.000 | 1.000 | 1.000 | 1.000 | 1.000 | 0.604 | 0.744 | 1.000 | 1.000 | 1.000 | 1.000 | 1.000 | 1.000 | 1.000 | 1.000 | 0.622 | 0.776 | 0.666 | 0.740 | 0.880 |
| | OF | NNCH | | | | | | | | | | | | | | | | 0.568 | 0.569 | 0.549 | 0.519 | 0.516 |
| | | Ours | | | | | | | | | | | | | | | | 0.647 | 0.655 | 0.572 | 0.534 | 0.528 |
| CS | CS | NNCH | | | | | | 0.601 | 0.576 | 0.539 | 0.520 | 0.515 | | | | | | 0.564 | 0.564 | 0.520 | 0.508 | 0.508 |
| | | Ours | | | | | | 0.604 | 0.578 | 0.550 | 0.520 | 0.519 | | | | | | 0.611 | 0.596 | 0.579 | 0.513 | 0.509 |
| | BS | NNCH | 0.732 | 0.728 | 0.751 | 0.734 | 0.708 | 0.943 | 0.945 | 0.943 | 0.929 | 0.916 | 0.682 | 0.611 | 0.649 | 0.610 | 0.708 | 0.928 | 0.909 | 0.814 | 0.586 | 0.588 |
| | | Ours | 0.820 | 0.821 | 0.849 | 0.787 | 0.810 | 0.952 | 0.967 | 0.966 | 0.963 | 0.936 | 0.764 | 0.780 | 0.819 | 0.779 | 0.763 | 0.982 | 0.967 | 0.951 | 0.856 | 0.764 |

Table A4: Full results of time steps required to reach 90% accuracy in the Label Concatenation toy experiment. We marked "-" for cases where none of the three seeds reached 0.9 accuracy by 100,000 steps.

| Level | Task | Method | RNN | | | | | Stack-RNN | | | | | Tape-RNN | | | | | LSTM | | | | |
|---|---|---|---|---|---|---|---|---|---|---|---|---|---|---|---|---|---|---|---|---|---|---|
| | | | 0.8 | 0.9 | 0.95 | 0.99 | 0.995 | 0.8 | 0.9 | 0.95 | 0.99 | 0.995 | 0.8 | 0.9 | 0.95 | 0.99 | 0.995 | 0.8 | 0.9 | 0.95 | 0.99 | 0.995 |
| R | EP | NNCH | 327 | 484 | 675 | 2337 | 3928 | 266 | 355 | 580 | 2020 | 3307 | 11471 | 3823 | 1772 | 5542 | 8103 | 1764 | 2895 | 4943 | 14404 | 21532 |
| | | Ours | 283 | 351 | 550 | 1939 | 3346 | 251 | 334 | 513 | 1614 | 2960 | 2141 | 818 | 1138 | 3731 | 5854 | 1092 | 2039 | 3520 | 10639 | 16072 |
| | MA | NNCH | 7097 | 10243 | 15182 | 45259 | 84269 | 7055 | 9785 | 14581 | 46549 | 90939 | 5831 | 8582 | 13699 | 47227 | 98821 | 18324 | 25063 | 35583 | 91993 | - |
| | | Ours | 5150 | 7954 | 12826 | 46954 | 96728 | 6037 | 9317 | 16293 | 72230 | - | 5234 | 7684 | 12171 | 44721 | 97804 | 9449 | 12997 | 20685 | 63424 | - |
| | PC | NNCH | 474 | 657 | 1038 | 2681 | 4624 | 588 | 867 | 1208 | 3513 | 4890 | 460 | 685 | 1059 | 2880 | 4203 | 5916 | 7249 | 8950 | 16497 | 27017 |
| | | Ours | 388 | 431 | 707 | 1859 | 3507 | 472 | 657 | 998 | 2620 | 4150 | 326 | 420 | 636 | 2134 | 3482 | 2894 | 3139 | 4187 | 9565 | 14677 |
| | CN | NNCH | 1854 | 2385 | 3759 | 10016 | 15868 | 1823 | 2505 | 3852 | 10448 | 17473 | 1426 | 2029 | 3202 | 9122 | 15113 | 21791 | 23597 | 27609 | 63541 | 94508 |
| | | Ours | 1249 | 1552 | 2352 | 7246 | 12186 | 1358 | 1881 | 2969 | 8865 | 14559 | 1111 | 1639 | 2520 | 7579 | 14497 | 8799 | 12034 | 16603 | 38170 | 54722 |
| CF | RS | NNCH | - | - | - | - | - | - | - | - | - | - | - | - | - | - | - | - | - | - | - | - |
| | | Ours | - | - | - | - | - | - | - | - | - | - | - | - | - | - | - | - | - | - | - | - |
| | MAB | NNCH | - | - | - | - | - | - | - | - | - | - | - | - | - | - | - | - | - | - | - | - |
| | | Ours | - | - | - | - | - | - | - | - | - | - | - | - | - | - | - | - | - | - | - | - |
| | SE | NNCH | - | - | - | - | - | - | - | - | - | - | - | - | - | - | - | 94270 | - | - | - | - |
| | | Ours | - | - | - | - | - | - | - | - | - | - | - | - | - | - | - | 64365 | 88420 | - | - | - |
| CS | MD | NNCH | 89917 | 43433 | 36993 | 16813 | 17049 | - | 67933 | - | 9955 | 26850 | 2422 | - | 39088 | 14961 | 33543 | - | 68997 | - | 74387 | 53723 |
| | | Ours | 2702 | 3375 | 3813 | 12901 | 21970 | - | - | 4072 | 11102 | 22919 | - | 3755 | 5608 | 10105 | 14660 | - | - | - | - | - |
| | OF | NNCH | - | - | - | - | - | - | - | - | - | - | - | - | - | - | - | - | - | - | - | - |
| | | Ours | - | - | - | - | - | - | - | - | - | - | - | - | - | - | - | - | - | - | - | - |
| | CS | NNCH | - | - | - | - | - | - | - | - | - | - | - | - | - | - | - | - | - | - | - | - |
| | | Ours | - | - | - | - | - | - | - | - | - | - | - | - | - | - | - | - | - | - | - | - |
| | BS | NNCH | - | - | - | - | - | 16971 | 16913 | 17950 | 69359 | 81300 | - | - | - | - | - | 65676 | 71221 | - | - | - |
| | | Ours | - | - | - | - | - | 4510 | 4425 | 4855 | 21356 | 61739 | - | - | - | - | - | 6403 | 5469 | 16603 | 74337 | 86417 |

Table A5: Full results of the best accuracy in the Input Length Confounding experiment.

| Level | Task | Method | RNN | | | | | Stack-RNN | | | | | Tape-RNN | | | | | LSTM | | | | |
|---|---|---|---|---|---|---|---|---|---|---|---|---|---|---|---|---|---|---|---|---|---|---|
| | | | 0.8 | 0.9 | 0.95 | 0.99 | 0.995 | 0.8 | 0.9 | 0.95 | 0.99 | 0.995 | 0.8 | 0.9 | 0.95 | 0.99 | 0.995 | 0.8 | 0.9 | 0.95 | 0.99 | 0.995 |
| R | EP | NNCH | 1.000 | 1.000 | 1.000 | 1.000 | 1.000 | 1.000 | 1.000 | 1.000 | 1.000 | 1.000 | 0.902 | 0.935 | 0.898 | 0.944 | 0.935 | 1.000 | 1.000 | 1.000 | 1.000 | 1.000 |
| | | Ours | 1.000 | 1.000 | 1.000 | 1.000 | 1.000 | 1.000 | 1.000 | 1.000 | 1.000 | 1.000 | 0.971 | 1.000 | 0.952 | 1.000 | 0.695 | 1.000 | 1.000 | 1.000 | 1.000 | 0.932 |
| | MA | NNCH | 1.000 | 1.000 | 1.000 | 0.281 | 0.220 | 1.000 | 1.000 | 1.000 | 0.366 | 0.220 | 0.927 | 0.470 | 0.526 | 0.220 | 0.220 | 1.000 | 1.000 | 1.000 | 0.220 | 0.219 |
| | | Ours | 1.000 | 1.000 | 1.000 | 0.221 | 0.219 | 1.000 | 1.000 | 1.000 | 0.220 | 0.220 | 1.000 | 0.494 | 0.481 | 0.374 | 0.220 | 1.000 | 1.000 | 1.000 | 0.221 | 0.220 |
| | PC | NNCH | 1.000 | 1.000 | 1.000 | 1.000 | 1.000 | 1.000 | 1.000 | 1.000 | 1.000 | 1.000 | 1.000 | 1.000 | 1.000 | 0.785 | 0.838 | 1.000 | 1.000 | 1.000 | 1.000 | 1.000 |
| | | Ours | 1.000 | 1.000 | 1.000 | 1.000 | 1.000 | 1.000 | 1.000 | 1.000 | 1.000 | 1.000 | 1.000 | 0.837 | 0.843 | 0.785 | 0.527 | 1.000 | 1.000 | 1.000 | 1.000 | 0.843 |
| | CN | NNCH | 1.000 | 1.000 | 1.000 | 1.000 | 1.000 | 1.000 | 1.000 | 1.000 | 1.000 | 1.000 | 0.999 | 1.000 | 0.999 | 1.000 | 0.942 | 1.000 | 0.796 | 0.696 | 0.220 | 0.220 |
| | | Ours | 1.000 | 1.000 | 1.000 | 1.000 | 1.000 | 1.000 | 1.000 | 1.000 | 1.000 | 1.000 | 1.000 | 0.999 | 0.916 | 0.922 | 1.000 | 0.917 | 0.878 | 0.798 | 0.427 | 0.220 |
| CF | MAB | NNCH | 0.357 | 0.302 | 0.262 | 0.210 | 0.208 | 0.357 | 0.288 | 0.243 | 0.203 | 0.219 | 0.273 | 0.250 | 0.243 | 0.223 | 0.205 | 0.521 | 0.321 | 0.313 | 0.296 | 0.281 |
| | | Ours | 0.329 | 0.282 | 0.276 | 0.203 | 0.238 | 0.292 | 0.252 | 0.267 | 0.202 | 0.223 | 0.281 | 0.313 | 0.308 | 0.200 | 0.201 | 0.576 | 0.382 | 0.312 | 0.310 | 0.316 |
| | SE | NNCH | 0.220 | 0.221 | 0.220 | 0.220 | 0.220 | 0.465 | 0.308 | 0.220 | 0.220 | 0.220 | 0.228 | 0.220 | 0.220 | 0.220 | 0.220 | 0.690 | 0.391 | 0.221 | 0.220 | 0.220 |
| | | Ours | 0.227 | 0.220 | 0.219 | 0.221 | 0.220 | 0.330 | 0.308 | 0.220 | 0.220 | 0.220 | 0.220 | 0.244 | 0.220 | 0.220 | 0.220 | 0.704 | 0.459 | 0.297 | 0.220 | 0.221 |
| CS | MD | NNCH | 0.531 | 0.688 | 0.688 | 0.685 | 0.527 | 0.684 | 0.634 | 0.731 | 0.532 | 0.685 | 0.531 | 0.527 | 0.531 | 0.530 | 0.527 | 0.533 | 0.530 | 0.533 | 0.529 | 0.527 |
| | | Ours | 0.533 | 0.528 | 0.527 | 0.527 | 0.527 | 0.668 | 0.623 | 0.608 | 0.643 | 0.527 | 0.529 | 0.529 | 0.528 | 0.538 | 0.685 | 0.573 | 0.586 | 0.842 | 1.000 | 0.998 |

Table A6: Full results of time steps required to reach 90% accuracy in the Input Length Confounding experiment. We marked "–" for cases where none of the three seeds reached 0.9 accuracy by 100,000 steps.

| Level | Task | Method | RNN | | | | | Stack-RNN | | | | | Tape-RNN | | | | | LSTM | | | | |
|---|---|---|---|---|---|---|---|---|---|---|---|---|---|---|---|---|---|---|---|---|---|---|
| | | | 0.8 | 0.9 | 0.95 | 0.99 | 0.995 | 0.8 | 0.9 | 0.95 | 0.99 | 0.995 | 0.8 | 0.9 | 0.95 | 0.99 | 0.995 | 0.8 | 0.9 | 0.95 | 0.99 | 0.995 |
| R | EP | NNCH | 309 | 370 | 521 | 1930 | 3800 | 309 | 448 | 755 | 2328 | 4018 | 68475 | 45216 | 37105 | 50194 | 43013 | 3005 | 5964 | 9335 | 20888 | 32947 |
| | | Ours | 296 | 356 | 466 | 2283 | 5293 | 311 | 464 | 834 | 2917 | 6759 | 14867 | 11722 | 34533 | 24669 | - | 3112 | 6621 | 10323 | 23087 | 55827 |
| | MA | NNCH | 12132 | 19205 | 28087 | - | - | 11434 | 19144 | 32116 | - | - | 43445 | 80522 | 75152 | - | - | 39481 | 54027 | 87417 | - | - |
| | | Ours | 10796 | 17877 | 31781 | - | - | 13177 | 23633 | 48375 | - | - | 11012 | 79071 | 80986 | - | - | 22819 | 34925 | 52669 | - | - |
| | PC | NNCH | 502 | 898 | 1427 | 3849 | 6838 | 578 | 855 | 2559 | 4691 | 11390 | 13079 | 2932 | 30664 | 13152 | 81872 | 9211 | 12674 | 16613 | 33169 | 35326 |
| | | Ours | 1494 | 4937 | 7598 | 15765 | 29675 | 639 | 1007 | 1572 | 6292 | 14069 | 14249 | 71712 | 59344 | - | - | 7180 | 11119 | 19473 | 47186 | 76303 |
| | CN | NNCH | 10435 | 5111 | 8804 | 23131 | 36916 | 6766 | 14594 | 8369 | 34854 | 40378 | 10942 | 24946 | 20889 | 34761 | 65703 | 99724 | - | - | - | - |
| | | Ours | 3368 | 5259 | 7881 | 24162 | 41335 | 4244 | 6444 | 9095 | 24517 | 42787 | 6675 | 10563 | 43188 | 54876 | 41833 | 75259 | - | - | - | - |
| CF | MAB | NNCH | - | - | - | - | - | - | - | - | - | - | - | - | - | - | - | - | - | - | - | - |
| | | Ours | - | - | - | - | - | - | - | - | - | - | - | - | - | - | - | - | - | - | - | - |
| | SE | NNCH | - | - | - | - | - | - | - | - | - | - | - | - | - | - | - | - | - | - | - | - |
| | | Ours | - | - | - | - | - | - | - | - | - | - | - | - | - | - | - | - | - | - | - | - |
| CS | MD | NNCH | - | 69826 | 70170 | 75080 | - | - | - | 70106 | - | 89150 | - | - | - | - | 86860 | - | - | 47866 | 50054 | 72405 |
| | | Ours | - | - | - | - | - | - | - | - | - | - | - | - | - | - | - | - | - | - | - | - |

Table A7: Influence of spurious correlations ([Left] Label Concatenation [Right] Input Length Confounding) with "ADD" method. BGF effectively addresses spurious correlation to a greater extent compared to Adam. However, the "ADD" method, which does not implement gradient normalization and balancing, appears to be highly susceptible to spurious correlation.

| $P =$ | 0.0 | 0.8 | 0.9 | 0.95 | 0.99 | 0.995 | | $P =$ | 0.0 | 0.8 | 0.9 | 0.95 | 0.99 | 0.995 |
|---|---|---|---|---|---|---|---|---|---|---|---|---|---|---|
| Adam | 0.809 | 0.804 | 0.793 | 0.772 | 0.733 | 0.708 | | Adam | 0.844 | 0.754 | 0.710 | 0.698 | 0.584 | 0.567 |
| BGF | 0.864 | 0.841 | 0.828 | 0.809 | 0.753 | 0.707 | | BGF | 0.910 | 0.755 | 0.712 | 0.701 | 0.595 | 0.560 |
| "ADD" | 0.839 | 0.818 | 0.804 | 0.771 | 0.676 | 0.617 | | "ADD" | 0.882 | 0.729 | 0.628 | 0.522 | 0.401 | 0.368 |

Table A8: Best Accuracy Comparison against ADD and BGF. Rows corresponding to the performance of BGF are highlighted in blue for improved visibility. Accuracies $> 90\%$, indicative of achieving OOD generalization, are marked in bold.

| Level | Task | Method | RNN | Stack-RNN | Tape-RNN | LSTM | TF (None) | TF (Sin-Cos) | TF (ALIBI) | TF (Rot) |
|---|---|---|---|---|---|---|---|---|---|---|
| R | EP | ADD | **1.000** | **1.000** | **1.000** | **1.000** | 0.530 | 0.627 | **0.916** | **1.000** |
| | | Ours | **1.000** | **1.000** | **1.000** | **1.000** | 0.533 | 0.683 | **0.967** | **1.000** |
| | MA | ADD | **1.000** | 1.000 | **1.000** | **1.000** | 0.226 | 0.225 | 0.243 | 0.238 |
| | | Ours | **1.000** | **1.000** | **1.000** | **1.000** | 0.227 | 0.223 | 0.252 | 0.248 |
| | PC | ADD | **1.000** | **1.000** | **1.000** | **1.000** | 0.544 | 0.530 | 0.530 | 0.530 |
| | | Ours | **1.000** | **1.000** | **1.000** | **1.000** | 0.556 | 0.533 | 0.532 | 0.541 |
| | CN | ADD | **1.000** | **1.000** | **1.000** | **1.000** | 0.891 | 0.851 | 0.845 | 0.565 |
| | | Ours | **1.000** | **1.000** | **1.000** | **1.000** | **0.979** | 0.349 | 0.871 | 0.566 |
| CF | SM | ADD | 0.630 | 0.765 | 0.813 | 0.727 | 0.510 | 0.529 | 0.654 | 0.601 |
| | | Ours | 0.577 | 0.763 | 0.826 | 0.718 | 0.510 | 0.530 | 0.624 | 0.605 |
| | RS | ADD | 0.734 | 0.736 | 0.726 | 0.710 | 0.542 | 0.541 | 0.738 | 0.720 |
| | | Ours | 0.761 | 0.715 | 0.731 | 0.707 | 0.542 | 0.542 | 0.728 | 0.708 |
| | MAB | ADD | 0.418 | **0.926** | 0.671 | 0.849 | 0.327 | 0.327 | 0.327 | 0.327 |
| | | Ours | 0.512 | **0.987** | **0.947** | **0.912** | 0.327 | 0.327 | 0.328 | 0.326 |
| | SE | ADD | 0.584 | 0.767 | 0.625 | **0.933** | 0.228 | 0.224 | 0.224 | 0.226 |
| | | Ours | 0.843 | **0.906** | 0.896 | **0.953** | 0.224 | 0.226 | 0.225 | 0.223 |
| CS | DS | ADD | 0.544 | 0.573 | 0.550 | 0.700 | 0.542 | 0.541 | 0.540 | 0.565 |
| | | Ours | 0.546 | 0.584 | 0.548 | 0.696 | 0.542 | 0.541 | 0.539 | 0.578 |
| | MD | ADD | **1.000** | **1.000** | **1.000** | 0.797 | 0.554 | 0.551 | 0.630 | 0.581 |
| | | Ours | **1.000** | 0.725 | **1.000** | 0.706 | 0.589 | 0.550 | 0.656 | 0.592 |
| | OF | ADD | 0.552 | 0.594 | 0.562 | 0.695 | 0.542 | 0.541 | 0.540 | 0.595 |
| | | Ours | 0.548 | 0.593 | 0.558 | 0.684 | 0.542 | 0.541 | 0.539 | 0.563 |
| | BA | ADD | 0.504 | 0.548 | 0.506 | 0.645 | 0.505 | 0.514 | 0.508 | 0.593 |
| | | Ours | 0.504 | 0.545 | 0.515 | 0.634 | 0.513 | 0.513 | 0.506 | 0.558 |
| | BM | ADD | 0.509 | 0.532 | 0.508 | 0.579 | 0.501 | 0.506 | 0.505 | 0.552 |
| | | Ours | 0.506 | 0.530 | 0.508 | 0.578 | 0.501 | 0.506 | 0.506 | 0.554 |
| | CS | ADD | 0.571 | 0.608 | 0.573 | 0.628 | 0.514 | 0.514 | 0.518 | 0.569 |
| | | Ours | 0.573 | 0.612 | 0.601 | 0.628 | 0.514 | 0.514 | 0.516 | 0.581 |
| | BS | ADD | 0.829 | **0.973** | 0.819 | **0.998** | 0.255 | 0.517 | 0.899 | **0.953** |
| | | Ours | 0.867 | **0.980** | 0.844 | **0.989** | 0.255 | 0.501 | **0.924** | **0.959** |

## A4 BGF WITHOUT NORMALIZATION

We now compare BGF to its more naïve version without the balancing and normalization steps. This "ADD" method is implemented as follows:

$$g_{\text{Add}_t} = \alpha * g_t + \beta * g_{\text{low}_t} \text{ where } (\alpha = 1 \ \& \ \beta > 0) \quad (A1)$$

Note that removing the $\alpha + \beta = 1.0$ condition no longer guarantees that $g_t$ and $g_{\text{low}_t}$ are properly balanced. We experiment with $\beta$ values of 0.5, 1.0, 2.0, and 3.0. The results of toy experiments are presented in Table A7. It reveals that while BGF consistently outperforms Adam overall, ADD consistently shows statistically significantly lower results compared to BGF (p-value ¡ 0.05). Moreover, as the degree of spurious correlation increases, ADD exhibits a noticeable decrease in performance, even compared to Adam. In Table A8, we compare the performance of ADD and BGF on AR tasks. ADD generally exhibited lower performance compared to BGF, and in many cases where BGF achieved an accuracy of 0.9, ADD failed to achieve this.

## A5   PERFORMANCE COMPARISON FOR ADAMW AND BGF

We demonstrated the performance of AdamW and BGF (implemented on AdamW for fairness) on recurrent models in Table A9. These results demonstrate that BGF clearly outperforms AdamW (with an average increase of 3.3%p and a maximum increase of 46.8%p) and successfully learns tasks that AdamW failed to learn completely.

Table A9: Best accuracy comparison against with AdamW. To guarantee experimental consistency and fairness, BGF was implemented on top of AdamW. Regular (R) tasks were excluded, as both AdamW and BGF achieved 1.0 on all of them. Accuracies > 90%, which are indicative of OOD generalization, are marked in bold.

| Arch. | Optim. | Context-Free (CF) | | | | Context-Sensitive (CS) | | | | | | |
|---|---|---|---|---|---|---|---|---|---|---|---|---|
| | | SM | RS | MAB | SE | DS | MD | OF | BA | BM | CS | BS |
| RNN | AdamW | 0.582 | 0.712 | 0.460 | 0.827 | 0.526 | **1.000** | 0.533 | 0.494 | 0.500 | 0.568 | 0.699 |
| | BGF | 0.569 | 0.747 | 0.538 | 0.817 | 0.537 | **1.000** | 0.527 | 0.498 | 0.500 | 0.572 | 0.885 |
| Stack-RNN | AdamW | 0.762 | 0.709 | 0.875 | 0.892 | 0.551 | 0.702 | 0.562 | 0.533 | 0.527 | 0.607 | **0.960** |
| | BGF | 0.762 | 0.715 | **0.982** | 0.880 | 0.554 | **1.000** | 0.561 | 0.526 | 0.523 | 0.610 | **0.962** |
| Tape-RNN | AdamW | 0.740 | 0.694 | **0.985** | **0.959** | 0.528 | 0.532 | 0.548 | 0.500 | 0.500 | 0.533 | 0.789 |
| | BGF | 0.827 | 0.703 | **0.995** | **0.923** | 0.539 | **1.000** | 0.541 | 0.510 | 0.500 | 0.537 | 0.759 |
| LSTM | AdamW | 0.717 | 0.697 | 0.856 | **0.940** | 0.667 | 0.645 | 0.635 | 0.635 | 0.544 | 0.625 | **0.972** |
| | BGF | 0.710 | 0.705 | **0.907** | **0.920** | 0.684 | 0.724 | 0.682 | 0.632 | 0.554 | 0.630 | **0.996** |

## A6   TRAINING DISTRIBUTION ACCURACY ANALYSIS

In typical ML problems, accuracy on the training distribution (IID) is known to be predictive of (Miller et al., 2021), or at least correlated with (Hendrycks et al., 2021), OOD generalization performance. In contrast, for inductive inference, even if a model perfectly learns the training distribution, it may learn shortcuts or favorable hypotheses depending on various inductive biases. In Table A10, we showed that the model's accuracy on the training distribution approaches 1.0, demonstrating that the AR task can indeed be considered an inductive inference problem. Reported average accuracy in the main: Adam 0.979, BGF 0.983 are averaged over all 15 AR tasks and 7 models (excluding Transformer without positional embedding).

## A7   TRAIN PERFORMANCE OF OTHER OPTIMIZERS

Table A11[a] continues from Table 2, presenting the remaining results. Optimization methods designed for OOD generalization in traditional vision tasks often fail to achieve high performance in the validation of AR tasks, and frequently show poor learning performance even on the training distribution.

## A8   EXTREME LENGTH GENERALIZATION ANALYSIS ON ADDITIONAL TASKS

Figure A1 presents the extreme generalization results up to length 1000 for each model across 9 different tasks. We excluded models that showed results close to random chance at a validation length of 100. In almost all experiments, BGF outperforms Adam in extreme OOD scenarios and maintains robust performance regardless of length when it has learned the general rule.

---

[a]The results in this table are run on as many random seeds ($1 \sim 3$) as possible. In the revised version, the updated table with results averaged over all three seeds will be included.

Table A10: Average **training distribution accuracy** of Adam and BGF optimizer. BGF rows are highlighted in blue. Accuracies > 90% are marked in bold.

| Level | Task | Optim. | RNN | Stack-RNN | Tape-RNN | LSTM | TF (None) | TF (Sin-Cos) | TF (ALIBI) | TF (Rot) |
|---|---|---|---|---|---|---|---|---|---|---|
| R | EP | Adam | **1.000** | **1.000** | **1.000** | **1.000** | 0.633 | **1.000** | **1.000** | **1.000** |
| | | BGF | **1.000** | **1.000** | **1.000** | **1.000** | 0.635 | **1.000** | **1.000** | **1.000** |
| | MA | Adam | **1.000** | **1.000** | **1.000** | **1.000** | 0.454 | **0.973** | **0.978** | **0.996** |
| | | BGF | **1.000** | **1.000** | **1.000** | **1.000** | 0.458 | **0.989** | **0.992** | **0.998** |
| | PC | Adam | **1.000** | **1.000** | **1.000** | **1.000** | **1.000** | **0.999** | **0.999** | **1.000** |
| | | BGF | **1.000** | **1.000** | **1.000** | **1.000** | **1.000** | **1.000** | **1.000** | **1.000** |
| | CN | Adam | **1.000** | **1.000** | **1.000** | **1.000** | **1.000** | **1.000** | **1.000** | **1.000** |
| | | BGF | **1.000** | **1.000** | **1.000** | **1.000** | **1.000** | **1.000** | **1.000** | **1.000** |
| CF | SM | Adam | **0.988** | **1.000** | **1.000** | **1.000** | 0.511 | **0.998** | **0.999** | **1.000** |
| | | BGF | **0.996** | **1.000** | **1.000** | **1.000** | 0.512 | **0.999** | **1.000** | **1.000** |
| | RS | Adam | **1.000** | **1.000** | **1.000** | **1.000** | 0.672 | **0.994** | **1.000** | **1.000** |
| | | BGF | **1.000** | **1.000** | **1.000** | **1.000** | 0.672 | **0.999** | **1.000** | **1.000** |
| | MAB | Adam | **0.978** | **0.997** | **0.981** | **1.000** | 0.610 | 0.618 | **0.956** | **0.982** |
| | | BGF | **0.983** | **1.000** | **0.995** | **1.000** | 0.610 | 0.769 | **0.957** | **0.988** |
| | SE | Adam | **0.999** | **1.000** | **1.000** | **1.000** | 0.588 | 0.716 | **0.993** | **0.991** |
| | | BGF | **0.999** | **1.000** | **1.000** | **1.000** | 0.592 | 0.706 | **0.999** | **0.998** |
| CS | DS | Adam | 0.873 | **0.954** | **0.975** | **1.000** | 0.672 | **0.998** | **1.000** | **1.000** |
| | | BGF | **0.902** | **0.963** | **0.982** | **1.000** | 0.672 | **0.999** | **1.000** | **1.000** |
| | MD | Adam | **1.000** | **1.000** | **1.000** | **1.000** | **1.000** | **1.000** | **1.000** | **1.000** |
| | | BGF | **1.000** | **1.000** | **1.000** | **1.000** | **1.000** | **1.000** | **1.000** | **1.000** |
| | OF | Adam | **0.938** | **0.986** | **0.997** | **1.000** | 0.672 | **1.000** | **0.993** | **1.000** |
| | | BGF | **0.949** | **0.991** | **0.999** | **1.000** | 0.671 | **1.000** | **0.997** | **1.000** |
| | BA | Adam | 0.893 | **0.943** | **0.959** | **1.000** | 0.445 | **0.990** | **0.998** | **1.000** |
| | | BGF | **0.911** | **0.953** | **0.962** | **0.999** | 0.444 | **0.995** | **0.999** | **1.000** |
| | BM | Adam | 0.840 | 0.893 | 0.886 | **0.945** | 0.554 | **0.942** | **0.942** | **0.962** |
| | | BGF | 0.851 | **0.902** | 0.893 | **0.964** | 0.552 | **0.956** | **0.951** | **0.970** |
| | CS | Adam | **0.937** | **0.972** | **0.940** | **0.981** | 0.644 | **0.968** | **0.956** | **0.965** |
| | | BGF | **0.941** | **0.972** | **0.958** | **0.983** | 0.644 | **0.971** | **0.957** | **0.968** |
| | BS | Adam | **0.995** | **1.000** | **1.000** | **1.000** | 0.452 | **1.000** | **1.000** | **1.000** |
| | | BGF | **0.999** | **1.000** | **1.000** | **1.000** | 0.451 | **1.000** | **1.000** | **1.000** |

Table A11: Result continued from Table 2. Other optimizers not only fail to achieve high performance in validation but also often exhibit poor learning performance even on the training distribution.

| Architecture | Optimizer | Train | | | | | Test | | | | |
|---|---|---|---|---|---|---|---|---|---|---|---|
| | | MAB | DS | OF | BM | CS | MAB | DS | OF | BM | CS |
| **RNN** | Adam | 0.978 | 0.873 | 0.938 | 0.840 | 0.937 | 0.458 | 0.541 | 0.544 | 0.503 | 0.569 |
| | SAM | 0.878 | 0.814 | 0.873 | 0.788 | 0.915 | 0.330 | 0.537 | 0.536 | 0.501 | 0.569 |
| | SWAD | 0.882 | 0.867 | 0.921 | 0.819 | 0.921 | 0.328 | 0.529 | 0.528 | 0.500 | 0.557 |
| | LPF-SGD | 0.878 | 0.863 | 0.920 | 0.812 | 0.918 | 0.329 | 0.527 | 0.526 | 0.503 | 0.554 |
| | BGF | 0.983 | 0.902 | 0.949 | 0.851 | 0.941 | 0.512 | 0.546 | 0.548 | 0.506 | 0.573 |
| **Stack-RNN** | Adam | 0.997 | 0.954 | 0.986 | 0.893 | 0.972 | 0.796 | 0.558 | 0.575 | 0.530 | 0.609 |
| | SAM | 0.908 | 0.867 | 0.921 | 0.835 | 0.941 | 0.390 | 0.563 | 0.564 | 0.509 | 0.589 |
| | SWAD | 0.898 | 0.905 | 0.950 | 0.838 | 0.944 | 0.339 | 0.545 | 0.562 | 0.507 | 0.586 |
| | LPF-SGD | 0.897 | 0.905 | 0.952 | 0.836 | 0.944 | 0.342 | 0.548 | 0.564 | 0.509 | 0.588 |
| | BGF | 1.000 | 0.963 | 0.991 | 0.902 | 0.972 | 0.987 | 0.584 | 0.593 | 0.530 | 0.612 |
| **Tape-RNN** | Adam | 0.981 | 0.975 | 0.997 | 0.886 | 0.940 | 0.508 | 0.541 | 0.541 | 0.502 | 0.569 |
| | SAM | 0.911 | 0.891 | 0.926 | 0.844 | 0.922 | 0.353 | 0.549 | 0.549 | 0.503 | 0.569 |
| | SWAD | 0.910 | 0.888 | 0.950 | 0.855 | 0.924 | 0.338 | 0.543 | 0.544 | 0.504 | 0.558 |
| | LPF-SGD | 0.913 | 0.886 | 0.954 | 0.853 | 0.921 | 0.340 | 0.538 | 0.545 | 0.504 | 0.558 |
| | BGF | 0.995 | 0.982 | 0.999 | 0.893 | 0.958 | 0.947 | 0.548 | 0.558 | 0.508 | 0.601 |
| **LSTM** | Adam | 1.000 | 1.000 | 1.000 | 0.945 | 0.981 | 0.820 | 0.650 | 0.560 | 0.551 | 0.618 |
| | SAM | 0.911 | 0.975 | 0.998 | 0.886 | 0.954 | 0.364 | 0.573 | 0.541 | 0.519 | 0.593 |
| | SWAD | 0.913 | 0.975 | 0.997 | 0.865 | 0.936 | 0.339 | 0.529 | 0.539 | 0.506 | 0.570 |
| | LPF-SGD | 0.925 | 0.972 | 0.998 | 0.866 | 0.936 | 0.339 | 0.531 | 0.530 | 0.514 | 0.570 |
| | BGF | 1.000 | 1.000 | 1.000 | 0.964 | 0.983 | 0.912 | 0.696 | 0.684 | 0.578 | 0.628 |

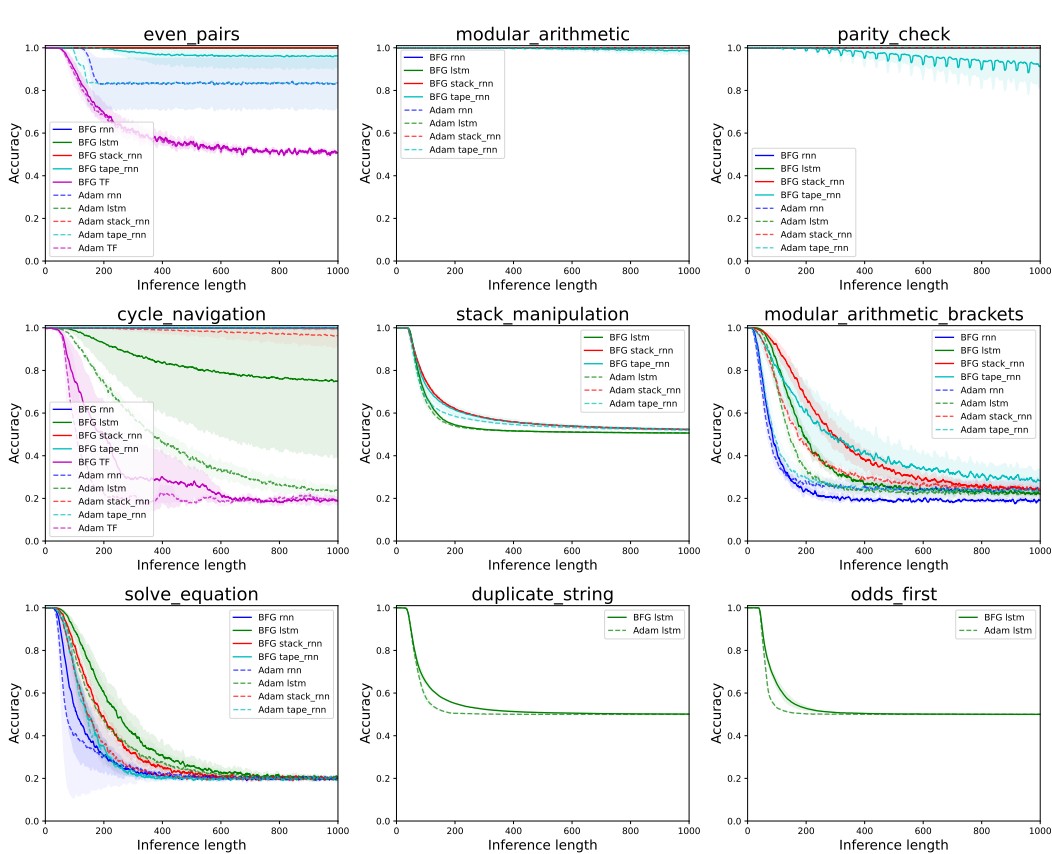

Figure A1: Additional results on length generalization on extreme OOD (up to length 1000). In each figure, the solid line represents the performance of BGF, while the dotted line represents the performance of Adam. The standard deviation across the 3 seeds is indicated by a shaded background of the same color. In most tasks, BGF achieves higher length generalization.

## R1 APPLICABILITY TO REAL-WORLD REASONING TASKS

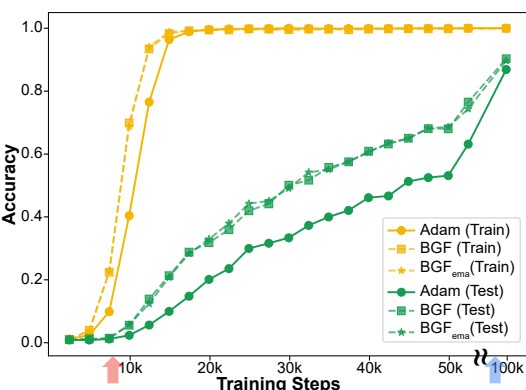

Figure R1: The training and test accuracy curves of the decoder-only transformer architecture with Adam, BGF, and BGF_ema on the composition task. The red and blue arrows denote the beginning of training and generalization phases.

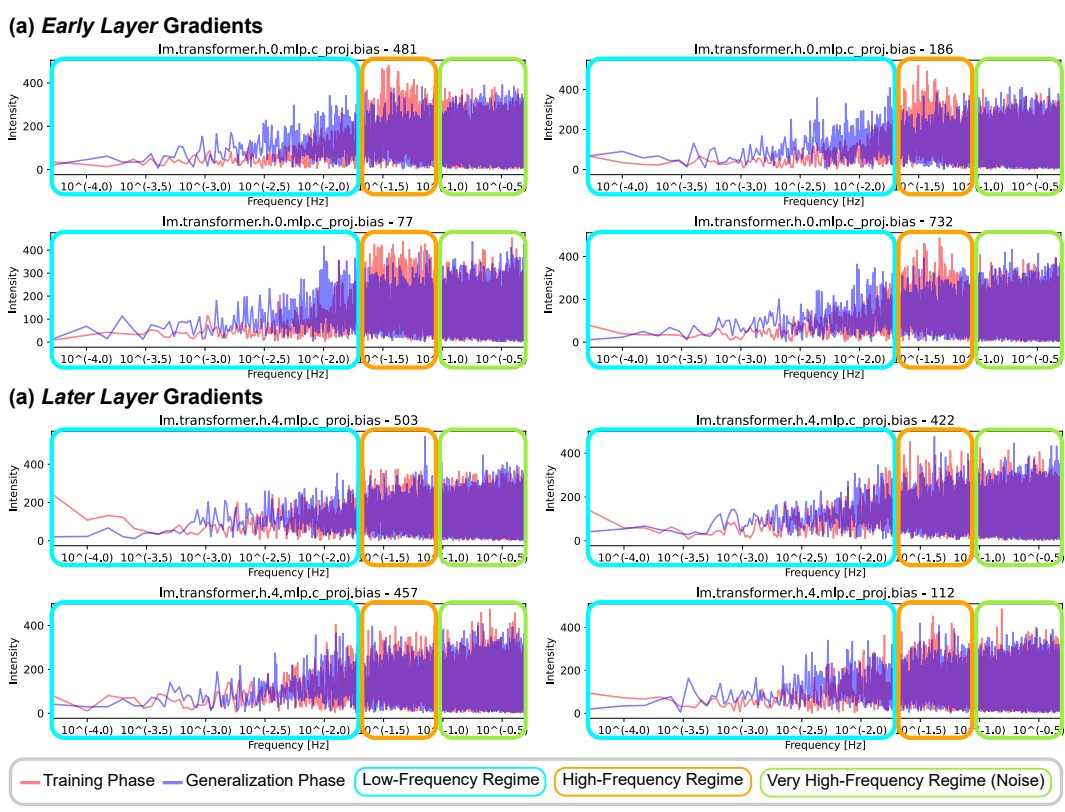

Figure R2: FFT results of training and generalization phase gradients obtained from (a) early and (b) later layers of the above decoder-only transformer architecture. Cyan, orange, and green boxes mark low-, high-, and very-high frequency regions. In early layers, the intensity of the generalization phase gradient is visibly dominant in the low-frequency regime, while in later layers, this difference diminishes.

## R2   LAYER-WISE GRADIENT ANALYSIS ON A DEEPER LSTM MODEL

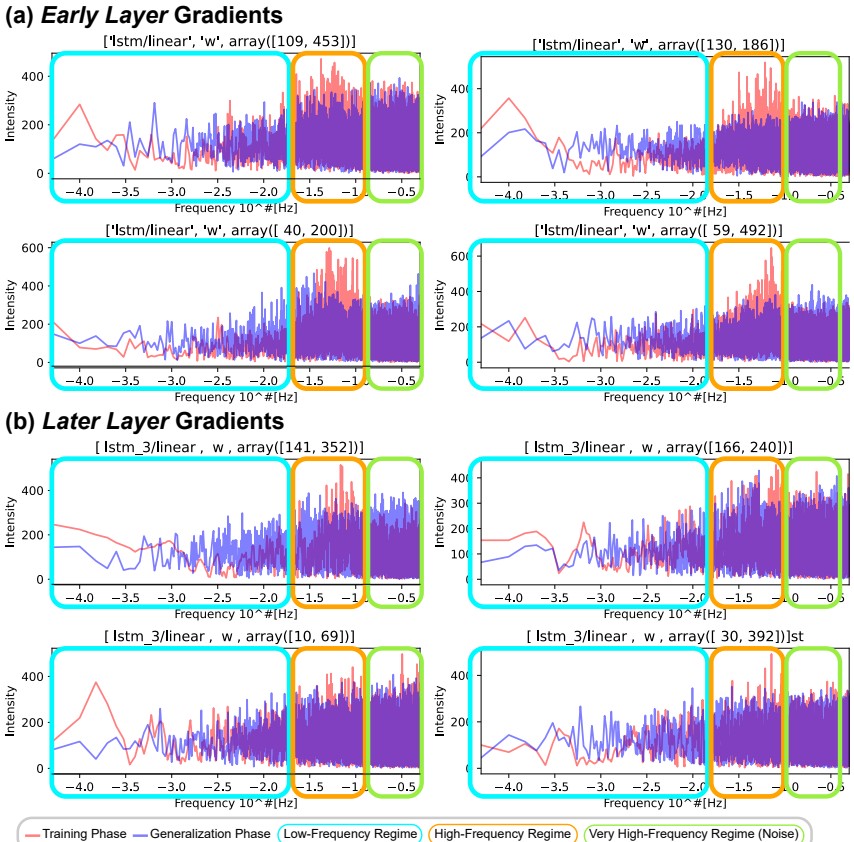

Figure R3: FFT results of training and generalization phase gradients obtained from (a) early and (b) later layers of the 4-layer LSTM architecture on the solve equation task. Cyan, orange, and green boxes mark low-, high-, and very-high frequency regions. In early layers, the intensity of the generalization phase gradient is visibly dominant in the low-frequency regime, while in later layers, this difference diminishes.

