# OpenReview forum: "Balancing Gradient Frequencies Facilitates Inductive Inference in Algorithmic Reasoning"
_ICLR.cc/2025/Conference — Submitted to ICLR 2025_

### Official Review · Reviewer_xigM · 2024-11-02

**Soundness:** 2
**Presentation:** 3
**Contribution:** 2
**Rating:** 3
**Confidence:** 5

**Summary:**

This paper introduces a new optimizer for Algorithmic reasoning tasks to avoid/mitigate shortcut learning for better OOD generalizability. Especially the work looks into the frequency domain of the gradients. The work provides toy experiments and synthesized spurious correlations in Algorithmic reasoning tasks.

**Strengths:**

•	If the claim is correct, the proposed optimizer would be simple and easy to implement, as it would be just a frequency (low/high) pass filter.

**Weaknesses:**

•	Most of all, the definition of shortcuts is the most serious problem that I have with this paper. I cannot agree with the authors’ definition of “shortcuts” which is stated as “non-generalizing hypotheses are termed “shortcuts.” Shortcut learning happens when a model can minimize the training loss just by looking at simpler features and giving up learning more complex core features. That means if such non-generalizing features are complex enough or more complex than core features, shortcut learning may not happen. That is why the name is ““short”cut” learning. That is why “non-generalizing hypotheses” cannot be just termed shortcuts.

•	Also, the sentence over lines 191-193 “Shortcuts can be interpreted as learning non-generalizing correlations between inputs and outputs that are valid in the training distribution but do not hold in the target distribution;” is not fully true. It depends on the target distribution. If the target distribution is still in distribution, such correlations will still be valid, but such correlations just won’t be visible.

•	This paper only considered two simple forms of synthetic spurious correlations on “toy” experiments. Algorithmic reasoning tasks are a good start and are used to examine the viability of an idea to apply to real-world tasks. It is unclear how this paper’s outcome can be useful for real-world tasks, vision and language. I am not confident to believe the claims stated in this paper as a general phenomenon. Also, nobody knows if something will go different in such Algorithmic reasoning tasks and conventional vision and language tasks?

•	For experiments with spurious correlations in Fig 2, the authors only experimented with spurious correlations of which portion is over 80% (when P >= 0.8). However, when the proportion is just that high the spurious correlation is trivial and straightforward.

**Questions:**

-	The following paper seems to have a good overlap with this submission:
“Frequency Shortcut Learning in Neural Networks,” Shunxin Wang, Raymond Veldhuis, Christoph Brune, and Nicola Strisciuglio, NeurIPS workshop on Distribution Shifts, 2022.
Hence, the novelty of this submission may be hurt.

-	For other points, please refer to the weaknesses.

---

> ### Author Response · Authors · 2024-11-18
> **1) Definition of shortcuts**
>
> Thank you for your thoughtful and detailed feedback. We appreciate the opportunity to clarify and strengthen the discussion around the definition of “shortcuts” in our paper.
>
> >**Regarding the definition of shortcuts**
>
> We acknowledge that “shortcuts” have been broadly defined in prior work, often in the context of classification tasks, as decision rules leveraging spurious correlations specific to a dataset, rather than semantically meaningful task-related cues. In AR, however, no exact definition of shortcuts currently exists, but previous works in AR *(What Algorithms can Transformers Learn? A Study in Length Generalization, ICLR ’24* / *Transformers Learn Shortcuts to Automata, ICLR ’22 Oral)* found that “shortcuts” correspond to non-generalizing hypotheses in reasoning tasks.
>
> As the reviewer notes, “shortcut learning” in classification tasks typically involve using easily learnable but dataset-specific features, rather than complex core features, due to the simplicity bias in Deep Neural Networks (DNNs). On the contrary, performing inductive inference to solve AR tasks requires models to generalize beyond the training distribution, distinguishing it from typical classification problems. The simplicity bias in AR usually occurs because DNNs learn non-generalizing hypotheses that are compatible with the training data but fail to extrapolate to out-of-distribution (OOD) tasks. Therefore, we opted to define the non-generalizing hypotheses as shortcuts specifically in the context of AR.
>
> Below, we clarify distinctions between AR and conventional classification problems that necessitate a separate definition of shortcuts for AR:
>
> 1. **Exhaustion of Training Data Information:** As mentioned in our paper, AR tasks often result in near-perfect training accuracy with the Adam optimizer, highlighting that the model utilizes all available information in the training data. However, the inductive leap required for OOD generalization depends on the ability to infer general logic rather than settle for non-generalizing patterns.
> 2. **The Role of Inductive Bias:** In AR, the challenge stems not from choosing between spurious and meaningful features but from selecting the “correct” generalization among infinite plausible hypotheses. For example, predicting the continuation of the sequence “1, 3, 5, 7” could yield numerous valid hypotheses, but general logic identifies “9” as the most parsimonious continuation.
> 3. **Non-Generalizing Hypotheses as Shortcuts:** In this context, we term “shortcuts” as those hypotheses that fail to generalize OOD, despite being valid within the training distribution. These shortcuts may not necessarily be simpler than the “general logic” but are misleading in the sense that they align with the training data but not the target task.
>
> >**On contents of lines 191–193**
>
> You pointed out that the statement *“Shortcuts can be interpreted as learning non-generalizing correlations between inputs and outputs that are valid in the training distribution but do not hold in the target distribution”* is not fully accurate as it depends on the target distribution. We agree with this nuance and appreciate the chance to refine our discussion. Our intention was to highlight that in AR tasks, the target distribution is always OOD (lines 30-32, 122-125, and 189-191) making non-generalizing correlations misleading for the intended task. However, as you rightly note, if the target distribution remains in-distribution, such correlations could still hold.
>
> >**Revised Definition Proposal**
>
> We sincerely apologize that the reviewer found our choice of terminology misleading, despite our efforts to draw a clear distinction between AR and conventional classification. Taking into account both your feedback and the general understanding of shortcuts in the literature, we propose refining our definition as follows:
>
> **Definition**: In algorithmic reasoning (AR), “shortcuts” refer to non-generalizing hypotheses that align with training data correlations but fail to extrapolate to out-of-distribution (OOD) tasks, which often define the target of AR problems. Unlike shortcuts in classification tasks—where they often involve simplicity bias favoring spurious correlations—shortcuts in AR tasks stem from the model’s inability to infer general logic among multiple plausible hypotheses fitting the training data.
>
> We believe this revision better aligns our work with the existing literature while addressing the unique challenges posed by AR tasks. We hope this explanation clarifies our perspective and makes our contributions to the understanding of shortcut learning more compelling.

---

> > ### Author Response · Authors · 2024-11-18
> > **2) Applicability to real-world tasks**
> >
> > Thank you for your thoughtful comment and for allowing us to discuss how our study could contribute to real-world tasks. Regarding the relevance of our work to real-world tasks, it is worth noting that even the state-of-the-art LLMs struggle with learning seemingly simple algorithms, which remains a critical challenge for LLMs.
> >
> > To demonstrate that BGF has the potential to address this critical limitation of Deep Neural Networks (DNNs), we verify its effectiveness on a real-world reasoning task with more complicated DNN architectures. We extended BGF to a decoder-only transformer structure and applied it to composition tasks—one of the most critical reasoning tasks in the real world (as discussed in *Grokked Transformers are Implicit Reasoners: A Mechanistic Journey to the Edge of Generalization, NeurIPS '24*). We observed that BGF significantly accelerates generalization on these tasks (please refer to Figure R1 of revised Appendix). Moreover, the additional gradient analysis of transformer layers (in Figure R2 of the revised Appendix) revealed a shift in the gradient frequency distribution during the training and generalization phases, as shown in Figure 3 of the main text. This finding suggests that BGF can enhance the reasoning capabilities of transform architectures. We hope that these additional results convince the reviewer that BGF does have practical, real-world applicability on more complex architectures and reasoning tasks.
> >
> > Furthermore, the ability to improving reasoning ability can help general-purpose LLMs overcome some of their weaknesses. A recent study, *Interpreting and Improving Large Language Models in Arithmetic Calculation (ICML '24 oral)*, found that LLMs exhibit separate attention heads for processing language and arithmetic, with these heads being largely decorrelated. By isolating the heads responsible for arithmetic and fine-tuning them on arithmetic datasets, the study demonstrated that LLMs can retain their language capabilities while significantly improving their mathematical reasoning. While it is infeasible to apply BGF to the above paper because the authors have not released their code base, the findings from this paper imply that using an optimizer specialized for algorithmic reasoning during logical head fine-tuning could improve the reasoning ability of LLMs while keeping their linguistic abilities intact.
> >
> > We are grateful for your comments, which allowed us to strengthen and expand the discussion around our work.

---

> > > ### Author Response · Authors · 2024-11-18
> > > **3) Purpose of toy experiments with synthetic spurious correlations**
> > >
> > > The purpose of our toy experiments was to demonstrate that Adam is indeed prone to spurious correlations in AR tasks, and that it tends to learn simpler, non-generalizing hypotheses (lines 198–199). While it is well-known that Adam favors simple, non-generalizing solutions in Vision and NLP, there has been little research on how Adam behaves on AR tasks. Through our experiments on four models across 11 tasks, we clearly showed that Adam exhibits the consistent tendency of learning such simple non-generalizing solutions in AR. Therefore, we believe our toy experiments are appropriately designed to study Adam’s properties on AR and successfully fulfills their purpose.

---

> > > > ### Author Response · Authors · 2024-11-18
> > > > **4) Spurious correlation experiments on  P < 80%**
> > > >
> > > > As per the reviewer’s request, we report additional results of the "Label Concatenation" toy experiment for P < 80%. Even at lower degrees of spurious correlation (P=0.4), Adam's performance on the bucket sort task decreases, and its training speed on the parity check task increases. Figure 2 will be revised to include results on lower degrees of spurious correlation.
> > > >
> > > > - Performance comparison on Bucket Sort
> > > >
> > > > |P=|0.0|0.4|0.5|0.6|0.7|
> > > > |-|-|-|-|-|-|
> > > > |Adam|0.831|0.828|0.813|0.826|0.824|
> > > > |BGF|0.877|0.903|0.898|0.886|0.889|
> > > >
> > > > - Training speed comparison on Parity Check
> > > >
> > > > |P=|0.0|0.4|0.5|0.6|0.7|
> > > > |-|-|-|-|-|-|
> > > > |Adam|939|1211|1302|1455|1599|
> > > > |BGF|711|764|817|850|875|

---

> ### Author Response · Authors · 2024-11-18
> **5) Comparison with the NeurIPS workshop paper**
>
> Thank you for introducing this interesting paper. However, we believe that the suggested paper and our work are different in multiple aspects.
>
> The paper you mentioned, *“Frequency Shortcut Learning in Neural Networks”*, focuses on analyzing input data (images) in the Fourier domain and identifying specific frequencies that act as shortcuts. Indeed, transforming input data into the frequency domain to analyze spurious correlations or components related to generalization is a well-established area of research, particularly in the vision domain (e.g., *High-frequency Component Helps Explain the Generalization of Convolutional Neural Networks, CVPR ‘20*).
>
> In contrast, our work takes a fundamentally different approach by linking the *frequency of gradients* to the optimization performance in algorithmic reasoning (AR) tasks. Specifically, we introduce temporal frequency filtering during the learning process to guide the model toward learning generalizing hypotheses. This is a novel perspective that has not been explored in prior work.
>
> Additionally, we highlight in Section 5.1 that optimizers like SAM, which have demonstrated strong generalization performance in vision tasks, perform poorly in AR tasks. This result further underscores the unique nature of AR tasks and the importance of studying optimization dynamics specifically tailored to these problems.
>
> We hope this clarifies the distinction between the contributions of our work and the referenced paper, and we are grateful for the opportunity to elaborate on these points.

---

> ### Comment · Reviewer_xigM · 2024-11-25
>
> Thank you for your response. However, I still have concerns with this paper.
>
> The authors’ new definition, **““shortcuts” refer to non-generalizing hypotheses that align with training data correlations but fail to extrapolate to out-of-distribution (OOD) tasks”** is not correct. There are multiple factors that hurt (OOD) generalization, and shortcut learning is one of them. That is, non-generalizing hypotheses can include shortcuts, but shortcuts solely cannot represent non-generalizing hypotheses. Also, if the authors would argue, “…..shortcuts may not necessarily be simpler than the “general logic.””, which is not true in classification tasks, they should not use the term “shortcut” learning in this different context which alters and misleads the well-established meaning in the research community and literature. I guess, with my best, what authors try to say is one type of bias, but not shortcut learning – even the authors emphasize that it is different from the one defined in classification tasks. My concern is that this altering terminology or misuse of terminology will bring a lot of confusion to the community. Also, this creates a fundamental doubt about the authors' understanding of the phenomenon.

---

> ### Author Response · Authors · 2024-11-25
>
> Thank you for your active engagement in the discussion period.
>
> Please refer to *Shortcut Learning in Deep Neural Networks,* Nature Machine Intelligence ‘20 [1]. Figure 3 states that “Among the solutions that solve the training data, only some generalize to an i.i.d. test set. Among those solutions, shortcuts fail to generalize to different data (o.o.d. test sets), but the intended solution does generalize.” Therefore, we believe our definition of shortcuts is not incorrect.
>
> Furthermore, the introduction of the above Nature paper states that “various phenomena [referring to learning under covariate shift, anti-causal learning, dataset bias, etc.] can be collectively termed shortcuts.” As stated in our initial response, the term “shortcuts” is flexibly used in AR [2,3,4], even if it does not strictly align with the the reviewer’s perspective. As such, we believe it is a stretch to argue that our use of the term “shortcuts” would alter or mislead its well-established meaning in the research community and literature.
>
> [1] Geirhos, Robert, et al. *"Shortcut learning in deep neural networks."* Nature Machine Intelligence ’20
>
> [2] Zhou, Yongchao, et al. *"Transformers Can Achieve Length Generalization But Not Robustly."*  ICLR ‘24 - Workshop
>
> [3] Liu, Bingbin, et al. *"Transformers Learn Shortcuts to Automata."* ICLR ‘22
>
> [4] Murty, Shikhar, et al. *"Pushdown Layers: Encoding Recursive Structure in Transformer Language Models."* EMNLP ‘23

---

> > ### Comment · Reviewer_xigM · 2024-12-03
> >
> > Thanks for the authors’ response.
> >
> > I make it clear that it is not a stretch. The authors stated that “These shortcuts may not necessarily be simpler than the “general logic”.” However, if a shortcut feature is simpler than the generalizable features/logic, it is not picked up, which is true in reasoning (both sequential and algorithmic) tasks, and vision and language tasks as well. Even the paper [1] that the authors referred to, clearly states that the depths of shortcuts are much smaller than the sequence length. To reiterate, that is why the name is shortcut learning. This notion is fundamentally too important for this paper but incorrectly understood throughout the entire paper.
> >
> > [1] Zhou, Yongchao, et al. "Transformers Can Achieve Length Generalization But Not Robustly." ICLR ‘24 - Workshop

---

> > > ### Author Response · Authors · 2024-12-04
> > >
> > > Thank you for your continued participation in the discussion period.
> > >
> > > First off, we agree with Reviewer xigM in that in Vision and Language domains, shortcuts generally entail simple and easy-to-learn features that result in non-generalizing hypotheses. And yet, through this rebuttal, we would like to point out that
> > > 1. our definition of “shortcuts” - non-generalizing hypotheses that fail on out-of-distribution data - is also widely-used and accepted in the research community; and
> > > 2. the term “shortcuts” is flexibly used to refer to spurious correlations or completely different concepts (as in the paper the reviewer referred to).
> > >
> > > Based on the above observations, we humbly ask the reviewer to be more accepting of our definition of “shortcuts” when evaluating our work.
> > >
> > > > **Our definition of “shortcuts” is widely-used and not misaligned with how it is used in the research community**
> > >
> > > In our paper: "Definition of Shortcuts: ...Predicting the next number in the sequence 1, 3, 5, 7 has infinite possible continuations. In this work, among hypotheses fitting the training data, those that enable OOD generalization (e.g., 9 for the continuation) are defined as “general logic”, whereas non-generalizing hypotheses are termed “shortcuts”. Shortcuts can be interpreted as learning non-generalizing correlations between inputs and outputs that are valid in the training distribution but do not hold in the target distribution.”
> > >
> > > Following are previous research works whose definition of shortcuts is in line with the above. None of them explicitly states that the simplicity of a solution is a necessary condition for shortcuts.:
> > > - “Shortcuts are decision rules that perform well on i.i.d. test data but fail on o.o.d. tests, revealing a mismatch between intended and learned solution.” - *Shortcut Learning in Deep Neural Networks,* Nature Machine Intelligence ‘20; (>2k citation)
> > > - ““shortcut features” – features which may be sufficient to solve a training task, but which may fail to generalize robustly and differ from the features preferred by people” - *What shapes feature representations? Exploring datasets, architectures, and training* NeurIPS ‘20
> > > - "We define shortcuts as statistical correlations in the data that allow a machine learning model to achieve high performance on a task without acquiring all the intended knowledge.” - *A Survey on Measuring and Mitigating Reasoning Shortcuts in Machine Reading Comprehension* arxiv ‘22
> > >
> > > In our original paper, we noted that the target distribution in Algorithmic Reasoning is by definition OOD. Therefore, we believe our definition is easily comprehensible and is not misaligned with how the term has previously been used in the community.

---

> > > > ### Author Response · Authors · 2024-12-04
> > > >
> > > > > **Flexible use of the term shortcuts**
> > > >
> > > > A. The following papers use the term “shortcuts” as spurious correlations at large:
> > > > - “Terms “spurious” and “shortcut” are largely synonymous in the literature, although the former often refers to features that arise unintentionally in a poorly constructed dataset, and the latter to features easily latched onto by a model.” - *On the Foundations of Shortcut Learning* ICLR ‘24 - Spotlight
> > > > - “Machine learning often achieves good average performance by exploiting unintended cues in the data. For instance, when backgrounds are spuriously correlated with objects, image classifiers learn background as a rule for object recognition. This phenomenon—called “shortcut learning”—at best suggests average metrics overstate model performance and at worst renders predictions unreliable as models are prone to costly mistakes on out-of-distribution (OOD) data where the shortcut is absent.” - *A Whac-A-Mole Dilemma : Shortcuts Come in Multiples Where Mitigating One Amplifies Others* CVPR ‘23
> > > > - “An increasing body of work has been conducted on understanding robustness in deep neural networks, particularly, how models sometimes might exploit spurious correlations and take shortcuts, leading to vulnerability in generalization to out-of-distribution data or adversarial examples in various NLP tasks such as NLI, Question-Answering, and Neural Machine Translation." *Identifying and Mitigating Spurious Correlations for Improving Robustness in NLP Models* NAACL-Findings ‘22
> > > >
> > > > B. In the paper *"Transformers Can Achieve Length Generalization But Not Robustly."* ICLR ‘24 - Workshop, the concept of “shortcut” is entirely different from your or our definition of shortcuts. From here on, we will refer to shortcuts in this specific paper as “shortcut (new).” According to this paper, “shortcut (new)” can be either non-generalizing solutions - for example, the solution used to solve the Dyck language task with the GPT model [2] - or intended/OOD-generalizing solutions - the solution for the Dyck language task learned by the Scratchpad+Recency model. The inclusion of the latter case means that “shortcut (new)” in this context is an entirely different solution that is different from both our and the reviewer’s suggested definition.
> > > >
> > > > Shortcut (new) denotes parallel processing of a given Algorithmic Reasoning task, which equates a transformer to automaton (in formal language theory) that processes inputs one by one as a logic processor. By definition, if a Shortcut (new) exists, and a transformer does indeed process Algorithmic Reasoning tasks in parallel, the depth of a transformer architecture involved in the computation, which is shortcut (new), is smaller than the sequence length.
> > > >
> > > > These works evidence that the term “shortcuts” is used in diverse contexts, particularly in Algorithmic Reasoning, as long as its scope is clearly defined within the paper.
> > > >
> > > > [1] Zhou, Yongchao, et al. *"Transformers Can Achieve Length Generalization But Not Robustly."* ICLR ‘24 - Workshop
> > > >
> > > > [2] Liu, Bingbin, et al. *"Transformers Learn Shortcuts to Automata."* ICLR ‘22-oral

---

### Official Review · Reviewer_cFMM · 2024-11-03

**Soundness:** 4
**Presentation:** 3
**Contribution:** 3
**Rating:** 8
**Confidence:** 4

**Summary:**

This study examines optimization’s impact on AR performance, showing through toy experiments that Adam, a commonly used optimizer, is susceptible to spurious correlations that hinder out-of-distribution (OOD) generalization. To address this, the authors introduce a novel optimization method, Balancing Gradient Frequencies (BGF), designed to reduce shortcut learning by balancing low- and high-frequency gradients, thus promoting stronger inductive inference. Extensive experiments across varied AR tasks and DNN architectures reveal that BGF significantly improves accuracy, accelerates convergence, and smoothens the loss landscape, enabling DNNs to tackle tasks previously deemed unmanageable.

**Strengths:**

1.	Through extensive experiments, the authors demonstrate that balancing gradient frequencies directly impacts the model’s ability to generalize beyond the training distribution. This insight is particularly valuable, as it highlights the frequency characteristics of gradients as a crucial factor in overcoming shortcut learning, a perspective that has been relatively unexplored in AR.
2.	The study documents a phenomenon termed "train-test splitting," in which DNNs achieve training accuracy early on but require additional steps to attain OOD test accuracy. This splitting behavior resembles the grokking phenomenon and is linked to changes in gradient frequency patterns. The discovery emphasizes the importance of low-frequency gradient components for OOD generalization, offering a new understanding of model behavior during training.
3.	BGF’s architecture-agnostic design is evidenced by improved performance across various DNN types, including RNNs, LSTMs, and Transformer models, on multiple AR tasks. This adaptability underscores BGF’s potential for broad application, marking it as a significant innovation in optimization for generalization across neural network architectures.

**Weaknesses:**

1. Although the manuscript compares BGF with optimizers like Adam, SAM, SWAD, and LPF-SGD, incorporating more recent advancements in optimization techniques that are known to enhance generalization would further strengthen the case for BGF's superiority.

2. Conduct a hyperparameter sensitivity analysis focusing on λ and other gradient-balancing factors (e.g., α and β). This experiment would provide insights into how these parameters influence BGF's ability to generalize, especially under varying data conditions.

3. To deepen insights into how BGF affects gradient dynamics, examining gradient frequencies across different layers of DNN architectures is recommended. This analysis could reveal whether specific layers benefit more from low-frequency gradients, enabling further fine-tuning of BGF for optimal performance in various model architectures.

**Questions:**

see above

---

> ### Author Response · Authors · 2024-11-18
>
> Thank you for your considerate comments and for recognizing the novel insights and broad applicability of BGF in improving generalization for AR tasks. As you mentioned, our work is the first to approach emerging AR tasks from the perspective of optimization. To reflect your comments, we added new comparison results between BGF and recent optimizers from ‘23 and ‘24 (More comparison with recent optimizers). These additional results further highlight the uniqueness of AR tasks and the superior performance of BGF in addressing their challenges. Furthermore, at the request of Reviewer xigM, we verified BGF on decoder-only transformers and a real-world “composition” task (Additional analyses and results). The success of BGF at improving the performance of decoder-only transformers on this real-world reasoning task indicates that BGF can indeed be used to improve the reasoning abilities of Deep Neural Networks.
>
> >**More comparison with recent optimizers**
>
> In Table 2 of the main paper, we compared BGF to SAM, SWAD, and LPF-SGD, and In Appendix 5, we compared BGF to AdamW. We additionally compare BGF to 2 recent optimization techniques: Marg-Ctrl loss and FSAM optimizer.
> - Marg-Ctrl Loss (NeurIPS '23) [1] : The authors of the Marg-Ctrl Loss paper observed that the default Empirical Risk Minimization loss prefers solutions that maximize the margin. Inspired by this observation, the Marg-Ctrl loss was developed to enforce uniform-margin solutions. It showed notable ability to mitigate shortcuts in **vision and language tasks**.  Below, we compare BGF to Marg-Ctrl loss. Because Marg-Ctrl loss was defined for a single-class classification problem, the comparative study is conducted on seven AR tasks with output length = 1. Like the rest of optimizers in the main paper, Marg-Ctrl falls short of BGF in all tasks.
>
> |||EP|MA|PC|CN|MAB|SE|MD|
> |-|-|-|-|-|-|-|-|-|
> |RNN|Marg-Ctrl|1.000|0.986|1.000|1.000|0.294|0.222|0.922|
> ||BGF|**1.000**|**1.000**|**1.000**|**1.000**|**0.512**|**0.843**|**1.000**|
> |S-RNN|Marg-Ctrl|1.000|1.000|1.000|1.000|0.306|0.327|0.532|
> ||BGF|**1.000**|**1.000**|**1.000**|**1.000**|**0.987**|**0.906**|**0.725**|
>
> - FSAM Optimizer (CVPR '24) [2]: The FSAM paper first decomposed the adversarial perturbations induced by SAM to analyze which gradient components play a crucial role in improving generalization. Their analysis revealed the importance of stochastic, rather than full, gradient noise components in promoting generalization, and thus, FSAM advances the original SAM optimizer by removing the full gradient noise components, which are detrimental to the mini-batch-wise optimization process. It demonstrates state-of-the-art generalization performance on the **vision task**. Additional comparative experiments on more seeds, architectures, and tasks are running and will be updated throughout the discussion period.
>   - The table below compares the average of best accuracies since not all seeds of the FSAM experiments have finished running yet.
>
> |||RS|SE|MD|BS|
> |-|-|-|-|-|-|
> |RNN|FSAM|0.701|0.224|1.000|0.380|
> ||BGF|**0.728**|**0.513**|**1.000**|**0.862**|
> |LSTM|FSAM|0.694|0.661|0.550|0.737|
> ||BGF|**0.704**|**0.926**|**0.646**|**0.984**|
>
> [1] Puli, Aahlad Manas, et al. *"Don’t blame dataset shift! shortcut learning due to gradients and cross entropy."*
>
> [2] Li, Tao, et al. *"Friendly sharpness-aware minimization."*

---

> ### Author Response · Authors · 2024-11-18
>
> > **Hyperparameter analysis on $\lambda$ and gradient balancing factors ($\alpha$ and $\beta$)**
> - Effect of changing $\lambda$: We used a fixed window size of 100 for all of our experiments. This value was chosen based on the gradient analysis shown in Figure 3, which demonstrated that the gradient patterns of both the training and generalization phases begin to change within the frequency range of 10^(-2)Hz and 10^(-1.5)Hz. A window size of 100 was appropriately chosen to suppress gradients in higher frequencies than this range.
>
>    In the table below, we study the effect of changing the smoothing factor of BGF-ema, which is equivalent to altering $\lambda$ of the original BGF. The larger the smoothing factor is, the larger $\lambda$ becomes. The results are reported in terms of the best accuracy averaged over tasks and 3 seeds. This ablation study was conducted BGF-ema instead of the original BGF due to the time constraint of the discussion period. According to the results, 0.98 appears to be the apposite smoothing factor. Also, BGF outperforms Adam at smoothing factors ranging from 0.95 to 0.99.
>
> ||Adam|BGF-ema w/ 0.95|w/ 0.98|w/ 0.99|w/ 0.995|w/ 0.998|
> |-|-|-|-|-|-|-|
> |RNN|0.530|0.591|**0.626**|0.589|0.601|0.569|
> |LSTM|0.733|0.773|**0.780**|0.778|0.738|0.708|
> |StackRNN|0.744|0.795|0.767|**0.796**|0.770|0.774|
> |TapeRNN|0.635|**0.771**|0.757|0.668|0.623|0.532|
>
> - Effect of changing gradient balancing factors: For [$\alpha$, $\beta$] combination, the following combinations were tested: [0.95, 0.05], [0.90, 0.10], [0.80, 0.20], [0.70, 0.30]. Hyperparameter sweep was conducted using sequences of lengths 50~59, equivalent to mild OOD sequences. We used 128 samples per sequence length. The average of results from a total of 1280 samples (=128 x 10) was used to determine the best hyperparameter for each optimizer. To evidence that the results on mild OOD sequences are indeed representative of performance on longer, more extreme OOD sequences, hyperparameter sweep results and the final test accuracy on sequence length=100 are juxtaposed below.
> 1. RNN
>     - Average validation accuracy on sequences of lengths 50 to 59
>     |Optim.|Acc.|
>     |-|-|
>     |Adam|0.852|
>     |BGF[0.95, 0.05]|0.800|
>     |BGF[0.90, 0.10]|0.829|
>     |BGF[0.80, 0.20]|0.903|
>     |BGF[0.70, 0.30]|**0.917**|
>     - Test accuracy on sequences of length=100
>     |Optim.|Acc.|
>     |-|-|
>     |Adam|0.738|
>     |BGF[0.95, 0.05]|0.734|
>     |BGF[0.90, 0.10]|0.804|
>     |BGF[0.80, 0.20]|0.831|
>     |BGF[0.70, 0.30]|**0.867**|
> 2. LSTM (Note that BGF yields 1.000 accuracy on two hyperparameter settings. In such cases, we selected the one that converged fatser to 0.9 according to the number of training steps)
>     - Average validation accuracy on sequences of lengths 50 to 59
>     |Optim.|Acc.|
>     |-|-|
>     |Adam|0.998|
>     |BGF[0.95, 0.05]|**1.000**|
>     |BGF[0.90, 0.10]|0.999|
>     |BGF[0.80, 0.20]|**1.000**|
>     |BGF[0.70, 0.30]|0.999|
>     - Test accuracy on sequences of length=100
>     |Optim.|Acc.|
>     |-|-|
>     |Adam|0.957|
>     |BGF[0.95, 0.05]|0.968|
>     |BGF[0.90, 0.10]|0.965|
>     |BGF[0.80, 0.20]|**0.989**|
>     |BGF[0.70, 0.30]|0.986|
>     - Number of training steps to 0.9
>     |Optim.|Acc.|
>     |-|-|
>     |Adam|1248|
>     |BGF[0.95, 0.05]|**911**|
>     |BGF[0.90, 0.10]|663|
>     |BGF[0.80, 0.20]|**707**|
>     |BGF[0.70, 0.30]|910|
>
> >**Gradient analysis on different layers of DNN architectures**
>
> - We used a 4-layer LSTM model to analyze the gradients from different DNN layers. This experiment was conducted on the solve equation task. Please refer to Figure R3 of the revised paper for the visualization results. In early layers (Figure R3 (a)), the training and generalization phase gradients show a clear split between the two, while in later layers (Figure R3 (b)), this gap between the two gradients is visibly reduced. From this result, we conjecture that early layers could benefit more from suppressing high-frequency gradients.
>
> - We additionally analyzed the gradients from different layers of the Transformer model on the composition task - one of the most critical reasoning tasks in the real world (as discussed in *Grokked Transformers are Implicit Reasoners: A Mechanistic Journey to the Edge of Generalization, NeurIPS '24*). Please refer to Figure R2 of the revised paper for the visualization results. We observe a similar tendency to the above in this experiment; the discrepancy between training and generalization phase gradients is visibly larger in early layer gradients. This result again implies that early layers could benefit more from suppressing high-frequency gradients.

---

> > ### Author Response · Authors · 2024-11-20
> >
> > >**Additional analyses and results (Pg 26-27 of the revised paper)**
> >
> > In the revised manuscript, we 1) additionally verified BGF on a decoder-only transformer using the composition task—one of the most critical reasoning tasks in the real world (as discussed in *Grokked Transformers are Implicit Reasoners: A Mechanistic Journey to the Edge of Generalization, NeurIPS '24*) (**Section R1 of the revised Appendix**), and 2) conducted a layer-wise frequency analysis of gradient signals (**Section R2 of the revised Appendix**). The first experiment, performed with the decoder-only transformer on the composition task, corroborates that BGF can be utilized to promote learning of generalizing rules in more complex architectures and real-world reasoning tasks. The gradient analysis on the first experiment again reveals the discrepancy in training and generalization phase gradient signals. The second analysis indicates that early layer gradients are more likely to benefit from BGF, providing a further insight into the mechanism of BGF.

---

> > > ### Author Response · Authors · 2024-11-25
> > > **Eagerly waiting for your response!**
> > >
> > > Dear Reviewer cFMM,
> > >
> > > Thank you for your positive feedback!
> > >
> > > In the provided author response, we
> > > - added additional comparison with recent optimizers and loss functions;
> > > - conducted additional hyperparameter search;
> > > - performed gradient analysis on different layers of deeper DNN architectures;
> > > - and verified the effectiveness of BGF on real-world reasoning task (composition task) for the decoder-only transformer.
> > >
> > > We understand that you are juggling a lot of papers and duties during the reviewing period.
> > >
> > > Yet, as we approach the end of the discussion period, we eagerly await to hear back from you.
> > >
> > > If you have any more questions or comments, please let us know at your earliest convenience.
> > >
> > > Sincerely, authors

---

### Official Review · Reviewer_oa4Z · 2024-11-03

**Soundness:** 3
**Presentation:** 3
**Contribution:** 3
**Rating:** 6
**Confidence:** 3

**Summary:**

This paper presents Balancing Gradient Frequencies (BGF), an optimizer that enhances inductive inference and OOD generalization in deep neural networks for algorithmic reasoning tasks. Unlike traditional optimizers like Adam, which can learn spurious correlations, BGF balances low- and high-frequency gradient components to avoid shortcut learning and improve OOD generalization. Experiments across various DNN architectures and AR tasks demonstrate BGF’s superior performance and faster convergence.

**Strengths:**

1. The method tackles improving OOD AR tasks by proposing a new optimizer, which is novel and interesting to me.

2. The empirical studies at the beginning of Section 3 are interesting and well-designed, effectively illustrating the susceptibility of standard optimizers like Adam to shortcut learning.

3. Extensive experiments show that BGF consistently outperforms Adam across multiple architectures and tasks.

**Weaknesses:**

1. The paper lacks formal theoretical analysis explaining why BGF outperforms traditional optimizers. While the empirical results are strong, a theoretical understanding of BGF’s benefits would enhance the contribution.

2. The study primarily compares BGF with Adam, with fewer comparisons to more recent optimizers designed to improve generalization. Including a wider range of baseline optimizers would strengthen the evaluation.

**Questions:**

Please refer to the weaknesses section.

---

> ### Author Response · Authors · 2024-11-18
>
> Thank you for your insightful comments and for recognizing the novelty and effectiveness of our approach in improving OOD generalization for AR tasks through a new optimizer. As you suggested, we added new comparison results between BGF and recent optimizers from ‘23 and ‘24 (**More comparison with recent optimizers**). These additional results further highlight the uniqueness of AR tasks and the superior performance of BGF in addressing their challenges. Furthermore, at the request of Reviewer xigM, we verified BGF on decoder-only transformers and a real-world “composition” task (**Additional analyses and results**). The success of BGF at improving the performance of decoder-only transformers on this real-world reasoning task indicates that BGF can indeed be used to improve the reasoning abilities of Deep Neural Networks.
>
> >**Theoretical analysis**
>
> We humbly agree with the reviewer in that our paper can be improved with a formal theoretical analysis of BGF. Unfortunately, Algorithmic Reasoning is an emerging and underexplored field in and of itself, and research on theoretical frameworks for analyzing the properties of AR remains to be done. We sincerely wished to accompany the proposed optimizer with more theoretical analyses, such as its convergence guarantees, but the lack of theoretical tools for studying AR left us with no other option but to strengthen our claim with extensive empirical evidence and analyses. We believe developing a theoretical framework to better understanding reasoning capabilities Deep Neural Networks would be an impactful future direction of research and will continue our endeavors to comprehend DNNs’ reasoning behaviors.
>
> >**More comparison with recent optimizers**
>
> In Table 2 of the main paper, we compared BGF to SAM, SWAD, and LPF-SGD, and In Appendix 5, we compared BGF to AdamW. We additionally compare BGF to 2 recent optimization techniques: Marg-Ctrl loss and FSAM optimizer.
> - Marg-Ctrl Loss (NeurIPS '23) [1] : The authors of the Marg-Ctrl Loss paper observed that the default Empirical Risk Minimization loss prefers solutions that maximize the margin. Inspired by this observation, the Marg-Ctrl loss was developed to enforce uniform-margin solutions. It showed notable ability to mitigate shortcuts in **vision and language tasks**.  Below, we compare BGF to Marg-Ctrl loss. Because Marg-Ctrl loss was defined for a single-class classification problem, the comparative study is conducted on seven AR tasks with output length = 1. Like the rest of optimizers in the main paper, Marg-Ctrl falls short of BGF in all tasks.
>
> |||EP|MA|PC|CN|MAB|SE|MD|
> |-|-|-|-|-|-|-|-|-|
> |RNN|Marg-Ctrl|1.000|0.986|1.000|1.000|0.294|0.222|0.922|
> ||BGF|**1.000**|**1.000**|**1.000**|**1.000**|**0.512**|**0.843**|**1.000**|
> |S-RNN|Marg-Ctrl|1.000|1.000|1.000|1.000|0.306|0.327|0.532|
> ||BGF|**1.000**|**1.000**|**1.000**|**1.000**|**0.987**|**0.906**|**0.725**|
>
> - FSAM Optimizer (CVPR '24) [2]: The FSAM paper first decomposed the adversarial perturbations induced by SAM to analyze which gradient components play a crucial role in improving generalization. Their analysis revealed the importance of stochastic, rather than full, gradient noise components in promoting generalization, and thus, FSAM advances the original SAM optimizer by removing the full gradient noise components, which are detrimental to the mini-batch-wise optimization process. It demonstrates state-of-the-art generalization performance on the **vision task**. Additional comparative experiments on more seeds, architectures, and tasks are running and will be updated throughout the discussion period.
>   - The table below compares the average of best accuracies since not all seeds of the FSAM experiments have finished running yet.
>
> |||RS|SE|MD|BS|
> |-|-|-|-|-|-|
> |RNN|FSAM|0.701|0.224|1.000|0.380|
> ||BGF|**0.728**|**0.513**|**1.000**|**0.862**|
> |LSTM|FSAM|0.694|0.661|0.550|0.737|
> ||BGF|**0.704**|**0.926**|**0.646**|**0.984**|
>
> [1] Puli, Aahlad Manas, et al. *"Don’t blame dataset shift! shortcut learning due to gradients and cross entropy."*
>
> [2] Li, Tao, et al. *"Friendly sharpness-aware minimization."*

---

> > ### Author Response · Authors · 2024-11-20
> >
> > >**Additional analyses and results (Pg 26-27 of the revised paper)**
> >
> > In the revised manuscript, we 1) additionally verified BGF on a decoder-only transformer using the composition task—one of the most critical reasoning tasks in the real world (as discussed in *Grokked Transformers are Implicit Reasoners: A Mechanistic Journey to the Edge of Generalization, NeurIPS '24*) (**Section R1 of the revised Appendix**), and 2) conducted a layer-wise frequency analysis of gradient signals (**Section R2 of the revised Appendix**). The first experiment, performed with the decoder-only transformer on the composition task, corroborates that BGF can be utilized to promote learning of generalizing rules in more complex architectures and real-world reasoning tasks. The gradient analysis on the first experiment again reveals the discrepancy in training and generalization phase gradient signals. The second analysis indicates that early layer gradients are more likely to benefit from BGF, providing a further insight into the mechanism of BGF.

---

> > > ### Author Response · Authors · 2024-11-25
> > > **Looking forward to your response!**
> > >
> > > Dear Reviewer oa4Z,
> > >
> > > Thank you for your positive feedback!
> > >
> > > In the provided author response, we
> > > - added additional comparison with recent optimizers and loss functions;
> > > - and verified the effectiveness of BGF on real-world reasoning task (composition task) for the decoder-only transformer.
> > >
> > > We understand that you are juggling a lot of papers and duties during the reviewing period.
> > >
> > > Yet, as we approach the end of the discussion period, we eagerly await to hear back from you.
> > >
> > > If you have any more questions or comments, please let us know at your earliest convenience.
> > >
> > > Sincerely, authors

---

> > ### Comment · Reviewer_oa4Z · 2024-11-25
> >
> > I appreciate the authors for their efforts and responses, which have addressed both of my major concerns (lacking theoretical groundings and more recent baselines). I am leaning towards acceptance and will keep my original score of 6.

---

> > > ### Author Response · Authors · 2024-11-26
> > >
> > > Thank you for your positive feedback.
> > > We are glad that we were able to address your major concerns through the rebuttal. We will make sure to incorporate the points you mentioned into the paper to strengthen it further.
> > > If you have any additional questions or suggestions to further improve the paper, we would be happy to hear them at any time.

---

### Official Review · Reviewer_peNt · 2024-11-04

**Soundness:** 3
**Presentation:** 3
**Contribution:** 3
**Rating:** 6
**Confidence:** 3

**Summary:**

This paper introduces a new optimization method called BGF (Balancing Gradient Frequencies), aimed at improving length generalization in  Algorithmic Reasoning (AR) through the mitigation of shortcut learning. Building on the experimental findings regarding Adam's vulnerability to spurious correlations in AR tasks, the authors propose that BGF mitigates these correlations by balancing high- and low-frequency gradients. This approach encourages the model to learn general rules rather than relying solely on patterns present in the training data. Extensive experiments are conducted, demonstrating the effectiveness of BGF across various DNN architectures and tasks.

**Strengths:**

Extensive experiments and analyses;

A novel approach to study the effect of optimization on generalization in AR tasks;

The results on multiple DNN architectures and tasks show that BGF effectively improves generalization on test data.

**Weaknesses:**

The task description in the main text lacks clarity. There are too many task-related descriptions in the appendix—such as the "Modular Arithmetic," "Solve Equation," and "Bucket Sort" tasks—that interrupt the flow of the main text and affect its readability.

**Questions:**

Since the low-pass filter is implemented using a moving average filter with a window size of \lambda, what is the basis for selecting the window size \lambda? How does \lambda value affect the results?

In Table 3 [Right], why is the result of gradient filtering with an EMA lower than that with a queue in the LSTM model?

---

> ### Author Response · Authors · 2024-11-18
>
> Thank you for your thoughtful comments and for recognizing the strength of our paper in proposing a novel optimization method for improving generalization in AR tasks. As you mentioned, our work is the first to approach emerging AR tasks from the perspective of optimization. Following your suggestion, we 1) clarified our task descriptions and interpretation of experimental results and 2) studied the effect of changing $\lambda$. During the rebuttal process, we also added new experimental results to demonstrate the unique and practical advantages of BGF. First, at the request of Reviewers oa4Z and cFMM, we added new comparison results between BGF and recent optimizers from ‘23 and ‘24, which further highlight the uniqueness of AR tasks and the efficacy of BGF in addressing their challenges (**Reviewers oa4Z, cFMM - More comparison with recent optimizers**). Second, at the request of Reviewer xigM, we verified BGF on decoder-only transformers for a real-world “composition” task (**Additional analyses and results**). The transferability of BGF to decoder-only transformers on this real-world task indicates that BGF can indeed be used to improve the reasoning abilities of Deep Neural Networks.
>
> >**Task descriptions**
>
> We apologize that task descriptions were not provided clearly in the main paper. We were trying to pack a lot of material within the 10-page constraint and had to move the detailed task descriptions to Appendix. In Pg. 4 of the revised paper, we included an example of how we plan to include task descriptions in the main paper (marked in red). Also, in Table A2 of the revised Appendix, we added a table with example input output pairs for each task to improve the readability of our work as a standalone paper.
>
> >**Effect of changing $\lambda$**
>
> We used a fixed window size of 100 for all of our experiments. This value was chosen based on the gradient analysis shown in Figure 3, which demonstrated that the gradient patterns of both the training and generalization phases begin to change within the frequency range of 10^(-2)Hz and 10^(-1.5)Hz. A window size of 100 was appropriately chosen to suppress gradients in higher frequencies than this range.
>
> In the table below, we study the effect of changing the smoothing factor of BGF-ema, which is equivalent to altering $\lambda$ of the original BGF. The larger the smoothing factor is, the larger $\lambda$ becomes. The results are reported in terms of the best accuracy averaged over tasks and 3 seeds. This ablation study was conducted BGF-ema instead of the original BGF due to the time constraint of the discussion period. According to the results, 0.98 appears to be the apposite smoothing factor. Also, BGF outperforms Adam at smoothing factors ranging from 0.95 to 0.99.
>
> ||Adam|BGF-ema w/ 0.95|w/ 0.98|w/ 0.99|w/ 0.995|w/ 0.998|
> |-|-|-|-|-|-|-|
> |RNN|0.530|0.591|**0.626**|0.589|0.601|0.569|
> |LSTM|0.733|0.773|**0.780**|0.778|0.738|0.708|
> |StackRNN|0.744|0.795|0.767|**0.796**|0.770|0.774|
> |TapeRNN|0.635|**0.771**|0.757|0.668|0.623|0.532|
>
> >**Why is the result of EMA lower than queue in Table 3 [Right]?**
>
> We note that on the LSTM model, BGF with queue and EMA both fail to achieve >90% accuracy, indicating that neither one of the implementations enabled LSTM to learn the generalizing logic of this task. Therefore, although BGF with queue attains slightly higher accuracy than that with EMA, it is difficult to ascertain that the queue-based implementation is superior to EMA-based implementation.
>
> >**Additional analyses and results (Pg 26-27 of the revised paper)**
>
> In the revised manuscript, we 1) additionally verified BGF on a decoder-only transformer using the composition task—one of the most critical reasoning tasks in the real world (as discussed in *Grokked Transformers are Implicit Reasoners: A Mechanistic Journey to the Edge of Generalization, NeurIPS '24*) (**Section R1 of the revised Appendix**), and 2) conducted a layer-wise frequency analysis of gradient signals (**Section R2 of the revised Appendix**). The first experiment, performed with the decoder-only transformer on the composition task, corroborates that BGF can be utilized to promote learning of generalizing rules in more complex architectures and real-world reasoning tasks. The gradient analysis on the first experiment again reveals the discrepancy in training and generalization phase gradient signals. The second analysis indicates that early layer gradients are more likely to benefit from BGF, providing a further insight into the mechanism of BGF.

---

> > ### Author Response · Authors · 2024-11-25
> > **Please let us know if you have any more comments!**
> >
> > Dear Reviewer peNt,
> >
> > Thank you for your positive feedback!
> >
> > In the provided author response, we
> > - studied the effect of changing the window size of BGF;
> > - added additional task descriptions and discussion of experimental results;
> > - and verified the effectiveness of BGF on real-world reasoning task (composition task) for the decoder-only transformer.
> >
> > We understand that you are juggling a lot of papers and duties during the reviewing period.
> >
> > Yet, as we approach the end of the discussion period, we eagerly await to hear back from you.
> >
> > If you have any more questions or comments, please let us know at your earliest convenience.
> >
> > Sincerely, authors

---

> > ### Comment · Reviewer_peNt · 2024-11-27
> >
> > Thanks to the authors for taking the time to conduct the additional experiments and provide expanded task descriptions.  These efforts have enhanced both the quality and readability of the paper. I am inclined towards acceptance and will maintain my original score.

---

> > > ### Author Response · Authors · 2024-11-28
> > >
> > > Thank you for your positive and encouraging comment!
> > >
> > > We are grateful that we had the chance to improve the quality and readability of our paper during the reviewing process. We will surely incorporate the material of our discussion in the revised paper. In the meantime, if you have any further questions, we would be happy to hear them.

---

### Author Response · Authors · 2024-12-02
**Overall Response**

We would first like to thank all of the reviewers and the area/program chairs for their participation in reviewing of our paper. We are fully aware that the reviewers and the area chair are juggling multiple papers during the reviewing period, and are immensely thankful for taking their time to read our author response.

Here, we provide a summary of our works’s strengths recognized by the reviewers and major discussion/experiments provided in the author response.

>**Strengths**
- **Reviewers peNt, oa4Z, and cFMM** all recognized that the study of optimization methods for Algorithmic Reasoning and the proposed optimization method are novel, well-designed, and effective.
- Most of the reviewers **[peNT, oa4Z, cfMM]** agreed that our analyses on the optimization behavior of Adam and the occurrence of “train-test splitting” offer valuable insights that illustrate the susceptibility of standard optimizers like Adam to shortcut learning and highlight a perspective that has been relatively unexplored in.
- The majority of reviewers **[peNT, oa4Z, cFMM]** commended the design and extensiveness of our experiments to demonstrate the efficacy of BGF, the proposed optimizer. In particular, **Reviewers peNT and cFMM** acknowledged that the architecture-agnostic applicability of BGF allows it to improve generalization performance across various types of DNNs.

>**Summary of Author Response**

During the author-reviewer discussion period, Reviewers peNT and oa4Z commented that our response enhanced the quality of our work and addressed their major concerns. In summary, our author response included following discussions and experiments:
- Extended hyperparameter search to study the effect of changing the window size of BGF ($\lambda$) and gradient balancing factors ($\alpha$ and $\beta$);
- More comparison with recent optimization methods for promoting generalization (Marg-Ctrl Loss and Friendly-SAM optimizer);
- Additional experiments on the decoder-only transformer / real-world composition task to verify BGF’s effectiveness on more complex architectures for solving real-world reasoning tasks;
- Gradient analysis on a deeper recurrent model and decoder-only transformer;
- Extended toy experiments on lower degrees of spurious correlation (P <80%); and
- Detailed task descriptions (in the Revised paper) and clarification on the definition of shortcuts.

Thanks to all the reviewers, all of the above material provided importance guidance to improve our paper.

---

### Meta-Review · Area_Chair_H6Jx · 2024-12-21

**Metareview:**

This is an interesting paper combining gradient analysis and neural algorithmic reasoning. To the best of my knowledge, this is the first time such a combination is studied; generally speaking, backward-pass analyses are rare in AR, which is a mistake considering how important learning dynamics can be to OOD generalisation. The Authors' analysis of gradient frequencies directly motivates a solution to balance them, mainly through a low-pass filtering procedure on the gradients observed by Adam.

I would say that in spite of its promising premises, three main challenges remain to the paper's acceptance:

* [xigM] Debate over whether the term "shortcut learning" is appropriately used;
* [oa4Z] The lack of rigorous motivation behind the proposed BGF method;
* [xigM] The downstream applicability of the results, as they are mainly performed over toy / synthetic tasks.

I'd like to make a note that the paper _does not_ need to satisfy all three of these to qualify as a successful ICLR paper, in my opinion. That being said, I would hope to see robust efforts being made to address at least two of them.

**On the treatment of shortcuts**

The key issue in the outstanding discussions seems to be on whether the concept of a shortcut must imply that the discovered solution is _simpler_ than the target solution. On this, I agree with _both_ the Authors and the Reviewer:

* Algorithmic Reasoning is a field where the term 'shortcut' has emerged and become commonplace, and I did not notice a clear requirement there that shortcut solutions need be simpler.
* The Reviewer calls out an important point: the term is commonly used outside of AR to imply something subtly but importantly different. Further, no clear definition of shortcuts is used jointly throughout prior AR papers, meaning it is not trivial to simply call upon prior work without discussing its nuances.

The right course of action would be to _modify the related work section_ to properly contextualise not only what the Authors' definition is, but _how it differs from previous instances where such a term is defined_.

It appears that the Authors made some proposals during the Rebuttal period but did not make any tangible attempts to modify the paper in this direction during revisions. Further, no deeper discussion of various AR definitions of shortcuts was proposed. Sadly, I cannot conclude with certainty that there have been robust steps towards reconciling and contrasting the definition against previous works.

**On rigorous motivations for BGF**

I acknowledge the Authors attempted to respond to this concern by Reviewer oa4Z by noting shortcomings in existing theoretical foundations of AR, through passages as follows:

* _"... research on theoretical frameworks for analyzing the properties of AR remains to be done"._
* _"... lack of theoretical tools for studying AR..."_

However, while the Reviewer acknowledged this, in my opinion the connection between gradient frequencies and shortcuts is currently tenuous at best: the argument is made on the basis of correlating gradient frequencies with a phase of "pure train" and "pure test" performance improvements. But I am unconvinced that there is a 1:1 correspondence between "pure test improvement" regime and "no shortcuts being learnt". Any test set, even if OOD, still only evaluates a particular distribution, and the model could be shortcutting for that particular distribution while neglecting other OOD splits.

For this reason, a stronger theoretical analysis linking gradient frequencies / BGF to shortcuts more thoroughly would have been very welcome.

Further, I'm afraid it is not true that theoretical AR research does not exist. Googling "Algorithmic reasoning" surfaces a Gradient blog post (https://thegradient.pub/neural-algorithmic-reasoning/) which has comprehensive references, many of them presenting theoretical frameworks:

[2] Xu et al. What Can Neural Networks Reason About?. ICLR’20.
[10] Xu et al. How Neural Networks Extrapolate: From Feedforward to Graph Neural Networks. ICLR’21.
[20] Bevilacqua et al. Size-invariant graph representations for graph classification extrapolations. ICML’21.
[22] Dudzik and Veličković. Graph Neural Networks are Dynamic Programmers. NeurIPS’22.

In particular, [10] should be relevant as it studies extrapolation capabilities of neural nets in the NTK regime.

I'm not saying the Authors could have directly applied any of these works, but at least discussing them and why they _wouldn't_ be applicable would have been valuable.

Overall, I see no signs of clear improvement on this front either.

Coupled with the fact no detailed experimentation was performed outside of AR-style synthetic tasks or tasks where specialised pre-training is performed, especially considering that BGF should be easily deployable on _any_ generic text fine-tuning dataset, lead to a recommendation of _rejection_ for this paper in its current form; none of the Reviewers opted to champion the paper.

**Additional Comments On Reviewer Discussion:**

Based on my reading of the Reviewers' discussion as well as the AC-Reviewer discussion with some of them afterwards, I would _strongly encourage_ the Authors to resubmit to a future venue, paying close attention to, in the very least:

* contrasting their definition of shortcuts against other established uses of the term, in a way inspired by their initial suggestion but taking substantially more care to contrast against prior AR works as well as prior works in classification;
* providing at least a more thorough discussion of prior theoretical works in AR foundations and why they (wouldn't) apply to the present work, and ideally make steps on simple theoretical links between gradient frequency and shortcuts. Note that this might require a more mathematically rigorous definition of 'shortcut', and this might overall be a good thing for the paper's clarity;
* trying to test out BGF on training datasets comprising generic natural language corpora (of scale which is accessible to the Authors, of course). Since BGF is a generic gradient optimisation technique, it should be applicable regardless of the choice of dataset, and its utility is worth validating outside of the strict OOD regime.

As described in my meta-review, I do not believe all three of these are necessary for this paper to pass the bar for a top-tier AI conference like ICLR. But, if meaningful strides are made towards all three, it would make the paper a certain clear accept in my opinion.

Let me end by stressing that I find the research direction presented in this paper to be very valuable to the AR community, and would really like to see the paper make a successful impact in the field. My hope is that with more robust foundations, clarity around the key concepts, and a more holistic evaluation, this can indeed be realised to its full potential!

---

### Decision · Program_Chairs · 2025-01-22

Reject